



**Characteristics of the main primary source profiles of particulate**
**matter across China: from 1987 to 2017**
Xiaohui Bi, Yuan Cheng, Qili Dai, Jianhui Wu, Jiaying Zhang, Yufen Zhang, Lu Wang,
Yingze Tian, Yinchang Feng[*]
State Environmental Protection Key Laboratory of Urban Ambient Air Particulate Matter
Pollution Prevention and Control, College of Environmental Science and Engineering, Nankai
University, Tianjin, 300350, China
[*]*Correspondence to*: Yinchang Feng (fengyc@nankai.edu.cn)
**Abstract**
Based on the published literatures and typical profiles from the source library of
Nankai University, a total of 3244 chemical profiles of the main primary sources of
ambient particulate matter across China from 1987 to 2017, including coal
combustion, industrial emissions, vehicle emissions, fugitive dust, biomass burning
and cooking emissions, were investigated and reviewed to trace the evolution of their
main components and identify the main influencing factors to the evolution. As a
result, the most complicated profiles are likely attributed to coal combustion and
industrial emissions, which are evidently influenced by the decontamination processes
and sampling techniques as well as the coal nature and the boiler types. The profiles
of vehicle emissions are dominated by OC and EC, and varied with the changing
standard of sulfur and additives in the gasoline and diesel as well as the sampling



methods. The profiles of fugitive dust, such as soil dust and road dust, are dominated
by the crustal materials and influenced by the sampling methods to some extent. The
profiles of biomass burning is impacted mainly by the biomass categories and
sampling methods. As expected, the profiles of cooking emissions is impacted mainly
by the cooking types and materials. The uncertainty analysis and cluster analysis of all
these source profiles are conducted to reveal the variations of the different source
profiles in the same source category and evaluate the differences between source
categories. A relatively large variation has been founded in the source profiles of coal
combustion, vehicle emissions, industry emissions and biomass burning, indicating
that it is necessary to establish the local profiles for these sources due to their high
uncertainties. While the profiles of road dust and soil dust present a less variation with
the stable chemical characteristics among the different profiles in the same category,
suggesting that the profiles of these sources could be referenced for the cities in China
when such local profiles are not available. The presented results highlight the need for
increasing investigation of more specific markers (e.g., isotopes, organic compounds
and gaseous precursors) beyond routine measured components to discriminate sources.
Additionally, specific focus should be placed on the sub-type of source profiles in the
future, especially for local industrial emissions in China, to support the air quality
research communities in their efforts to develop high resolution source apportionment
for making a more effective control strategies.
*Keywords*: Source profiles; particulate matter; source apportionment.



## 1. Introduction

In light of preventing us from being exposure to high level of PM, source apportionment technique is a critical tool to help us in quantitative recognition of the source contributions of ambient particulate matter (PM) and developing efficient and cost-effective abatement policy. Source profile is of great importance in the application of receptor models for source apportionment study as it characterizing specific sources from the chemical point of view that revealing the signatures of source emissions (Hopke, 2016). Note that the measurement of source samples is costly and tough. Therefore, tons of studies using factor analytical model (source-unknown models, such as PMF, PCA etc) instead of using chemical mass balance (CMB) model (source profiles need to be known *a priori*) for the source apportionment. However, source sampling is essentially a very important basic work to get to know source signature and then make source identification and apportionment possible. In addition to source apportionment study, source profiles have played an important role in calculating source-specific emissions of individual compounds and converting total emissions from sources into the speciated emissions for air quality models, which can further provide effective strategies for environmental management (Simon et al., 2010).

In the past decades, source profiles of particulate matter from a variety of source types were substantially developed all over the world, especially in USA (Simon et al., 2010), Europe (Pernigotti et al., 2016) and East Asia (Liu et al., 2017). Most of the source profiles in China can be roughly divided into coal combustion (CC), industrial



emissions (IE), vehicle emissions (VE), fugitive dust (FD), biomass burning (BB),
cooking emissions (CE) etc. These available profiles have filled the gap of the
knowledge of source compositions and provided effective markers for the source
apportionment studies. With the development of sampling and chemical analysis
techniques, more valuable information, such as organic compounds, isotopes and size
distribution etc., has been explored to further expand the existing or new profiles. The
new valuable information gives significant possibilities to source apportionment
models to get more precise and reliable result.
From 1980s, source profile studies were initially implemented in China (Dai et al.,
1987) . During the past three decades, hundreds of source profiles have been achieved
across China. These profiles covered more than forty cities and several source types.
Source measurement is actually It is time to overview these source profiles along the
time line and give more profile knowledge to the atmospheric research community.
This review is based on the following ideas. In Section 2.1, we summarized the types
and the number of particulate source profiles in China published since the 1980s, and
reviewed the technological innovations of the sampling and chemical analytical
methods for source samples. In Section 2.2, we discussed the characteristics and
evolutions of source profiles including coal combustions, industrial emissions, vehicle
emissions, fugitive dust (soil dust and road dust), biomass burning and cooking
emissions. We also investigated the effect of various impact factors on source profiles.
In section 2.3, we used the coefficient of variation (CV, the standard deviation divided
by the mean) to further characterize the homogeneity of sources within the same



source category. Moreover, we also explored the heterogeneity between different
source categories through cluster analysis. In Section 3, we summarized the main
findings and a few issues of current source profiles, as well as the future requirements
for the development of source profiles in China.

## 2.   Overview of source profiles across China

We have used the key words included source profile/chemical profile/source
emissions, source apportionments/source contributions/particulate matter, and China
for literature research, searching for papers and dissertations in Chinese on China
National Knowledge Infrastructure (CNKI) and papers in English on Elsevier
ScienceDirect, respectively. The source profile data were compiled. After literature
searching (peer-reviewed papers published in international and Chinese journals), a
total of 374 published source profiles since 1980s across China were collected. In
general, all of these profiles were eventually divided into six source categories, with
70 of them were attributed to coal combustion, 32 to industrial emissions, 33 to
vehicle emissions, 118 to fugitive dust, 32 to cooking emissions, and 89 to biomass
burning. For the certain aerodynamic size, it obtained a total of 230 $PM_{2.5}$ profiles,
112 $PM_{10}$ profiles, and 32 for other sizes. The overview of these profiles are shown in
Fig. 1.





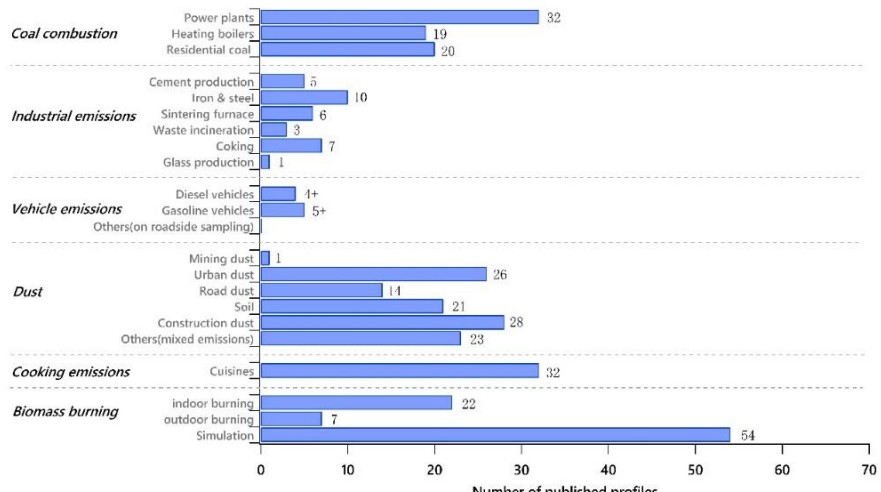


**Figure 1.** Overview of published source profiles across China.


In fact, more profiles measured in real-world in China were actually not published. A
database of source profiles in China founded by Nankai University contains 2870
profiles across China. In this paper, the characteristics of the published primary
profiles and some typical profiles of particulate matter founded by Nankai University
were discussed.

**2.1 Development of sampling and analysis techniques**
In the past thirty years, the sampling and chemical analysis techniques used in the
source apportionment research in China have been significantly improved to catch the
emissions of particles from various complex sources in real-world. In 1980s, the
samples from different sources were mainly obtained by sampling the dust directly
from the precipitators (CC, et al.), or the surface of fugitive dust sources (soil, road



dust, et al.) (Dai et al., 1987;Qu, 2013). Apparently, such sampling method cannot
catch the real emissions from the sources to the ambient air, especially for the CC or
other emission sources with humid and high-heat fume. The compositions of the
particulate matter in such fume will be changed due to the chemical reactions during
their dispersion process in the ambient air. With the development of sampling
techniques, samples were obtained that could reflect the real compositions from the
sources in 2000s (Hildemann et al., 1989;Lind et al., 2003;Ferge et al., 2004;Wang et
al., 2012). Nowadays, such technique called dilution tunnel sampling has been widely
used in China (Li et al., 2009). In the published profiles, 29% coal combustions, 91%
industrial emissions, and 12% biomass burning profiles were obtained with dilution
tunnel sampling method (as shown in Fig. 2).
Another problem is how to get the particle samples with certain aerodynamic size
from the sources of fugitive dust. In 1980s-1990s, the Bacho particle size analyzer
was widely used to obtain the size distributions from the source samples (Kauppinen
et al., 1991). Due to the low efficiency and potential safety risk of Bacho sampler, a
new sampling technique called the resuspended chamber was developed in 1990s by
Chow et al. (1994), and has been widely used since 2000 in China. This method could
obtain the particle sample with the certain aerodynamic size from the dust powder
collected from the source field. Nowadays, most source samples with the particle
aerodynamic size of 2.5 μm or 10 μm of fugitive dust were collected by the
resuspended sampling method in China (Ho et al., 2003;Zhao et al., 2006). Although
the resuspended chamber couldn't completely simulate the real environment, it still is





the best choice for the collection of fugitive dust samples until now.

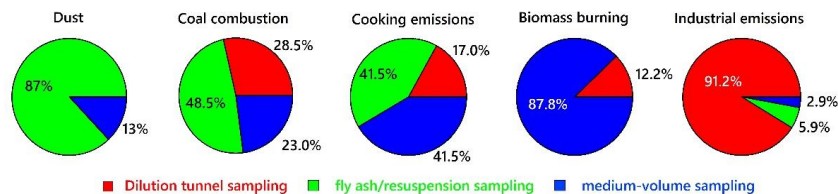


**Figure 2.** Share of sampling methods for the collection of source samples in China from
literatures.

The chemical analysis methods have been significantly improved since 1980s. A
typical source profile usually contains elements (e.g., Al, As, Ca, Cd, Cr, Cu, Fe, K,
Mg, Mn, Na, Pb and Zn), organic carbon (OC), elemental carbon (EC), and
water-soluble ions (WSI, e.g., $Cl^-$, $NO_3^-$, $SO_4^{2-}$, $NH_4^+$, $K^+$, $Na^+$, $Mg^{2+}$ and $Ca^{2+}$) in
China. Details procedures in terms of the establishment of different source profiles are
available in previous publications (Chow et al., 1994;Chow et al., 2004;Hou et al.,
2008b;Pei et al., 2016).
Teflon-membrane filters were analyzed for elements major by Inductively Coupled
Plasma Optical Emission Spectrometer (ICP-OES) or Inductively Coupled Plasma
Atomic Emission Spectrometer (ICP-AES). In recent years, Inductively Coupled
Plasma Mass Spectrometry (ICP-MS) and X Ray Fluorescence were also used, with
measurement systems have lower threshold and higher accuracy (Tsai et al., 2004).
For thermal/optical carbon analyzer, DRI Model 2001A and Sunset-Lab are the most
widespread technique, were used to analyze organic carbon and elemental carbon on
quartz filter by the thermal/optical reflectance (TOR) method (Chow et al., 1994;Ho
et al., 2003;Chow et al., 2004;Zhang et al., 2007) in source samples. Quartz fiber





filters were normally used for the determination of WSI by different types of Ion
Chromatography (IC) with high-capacity cation-exchange column and
anion-exchange column (Qi et al., 2015).
Initially, the mass balance models were developed for specific elements and particular
source types in 1970s (Winchester and Nifong, 1971;Miller et al., 1972). To improve
the discrimination of sources, more chemical species were subsequently introduced
into receptor models. Tracer species, a unique species that can be used as an indicator
of a particular source, playing an important role in estimating source contributions.
However, most of the source profiles in China are constituted of inorganic species,
with only a few studies providing information of organic compounds. Organic tracers
are of great value in source apportionment studies, as it is more source-specific than
inorganic species. For example, leveglucosan is a well-known organic tracer
represents for biomass burning (Lee et al., 2008), azzaarenes as a marker of inefficient
coal combustion (Junninen et al., 2009;Bi et al., 2008),sterols, monosaccharide
anhydrides and amides as a marker of cooking emissions (Schauer et al., 1999;Cheng
et al., 2016;Schauer et al., 2002;He et al., 2004;Zhao et al., 2007b, a).
The VOCs source profiles from China have been measured from various emission
sources by measuring both hydrocarbons and oxygenated VOCs (OVOCs) (Mo et al.,
2016). And the analysis of 16 USEPA priority PAHs was performed using a gas
chromatograph coupled with a mass spectrometer (GC-MS) to determine source
profile species (Cai et al., 2017a). Furthermore, for better discriminating sources, Pb
stable isotopes, which are not obviously influenced by ordinary chemical, physical or
biological fractionation processes (Gallon et al., 2005;Cheng and Hu, 2010), were
determined with an ICP-MS. Additionally, some other isotope measurements, for
example radiocarbon (Wang et al., 2017), sulfur (Han et al., 2016), and nitrogen (Pan



et al., 2016), as well as natural silicon (Lu et al., 2018), have also been reported to be
used as source indicators recently.
The above efforts indicate that the reported source profiles were collected by different
sampling methods and analyzed by different instruments, making the source profiles a
high uncertainty of comparability. Thus it is very urgent to establish standards for the
procedures of source sampling, chemical analysis and QA/QC to ensure the
representativeness, validation and comparability of source profiles in China.

**2.2  Characteristics and evolution of source profiles**
**2.2.1 Coal combustion and Industrial emissions**
As the most complicated source types, the source profiles of CC are influenced by
several factors, such as coal nature, boiler type, decontamination devices etc. Despite
all the influencing factors, the source profiles of CC in China are mainly consisted of
crustal materials, OC, EC, $SO_4^{2-}$ and trace metals. There are great differences in the
source profiles from different CC sources. Fig. 3 shows difference of the chemical
composition of source profiles between industrial boilers with wet desulfurization
(IBW) and power plant boilers with wet desulfurization (PPW) using the same
sampling method, as Mg, Al, Si, Ca, $SO_4^{2-}$, $NH_4^+$ and OC in the profile of PPW are
higher than that of IBW, while $NO_3^-$, $K^+$ and $Mg^{2+}$ are lower.





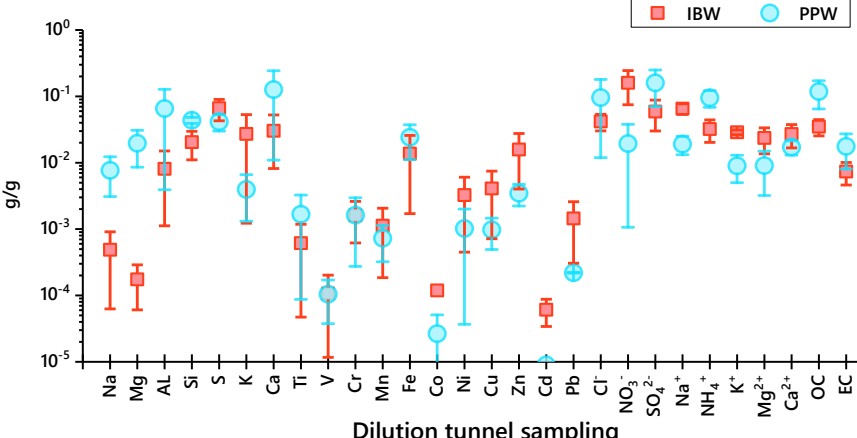


**Figure 3.** Compositions of source profiles between different boiler categories. IBW and PPW
denote industrial boilers using wet desulfurization and power plant boilers using wet
desulfurization, respectively. Data were collected from the source library of Nankai
University.

Within the same sampling method (dilution tunnel sampling method) and the same
boiler category, the characteristics of the source profiles of coal-fired power plants
equipped with different dust removal and desulfurization facilities are compared (Fig.
4). OC and EC in the profiles of the electrostatic precipitators (EP) are higher than
that in the electric bag compound dust collectors (EBCC), with average values of
0.1182±0.1254 and 0.0175±0.0196 g/g, respectively. High Ca in the source profiles
obtained by the electric bag compound dust collector is found as well (0.2307±0.0491
g/g).
Comparing data from different desulfurization facilities (Fig. 4), $SO_4^{2-}$ and Ca in
PM$_{2.5}$ profiles from the wet flue gas desulfurization (WFGD) is much higher than that





from dry desulfurization (DD). It suggested that $SO_4^{2-}$ is converted from $SO_2$ in the
flue gas through a limestone slurry washing reaction and then discharged with the
fume (Ma et al., 2015). Ca is also infused in the fume when the flue gas went through
the limestone washing process. OC in $PM_{2.5}$ profiles from the WFGD is also higher
than that from DD, indicating that the conversion of gaseous organics to the
particulate state caused by the wet desulfurization consequently increase the OC
content (Chen et al., 2005).

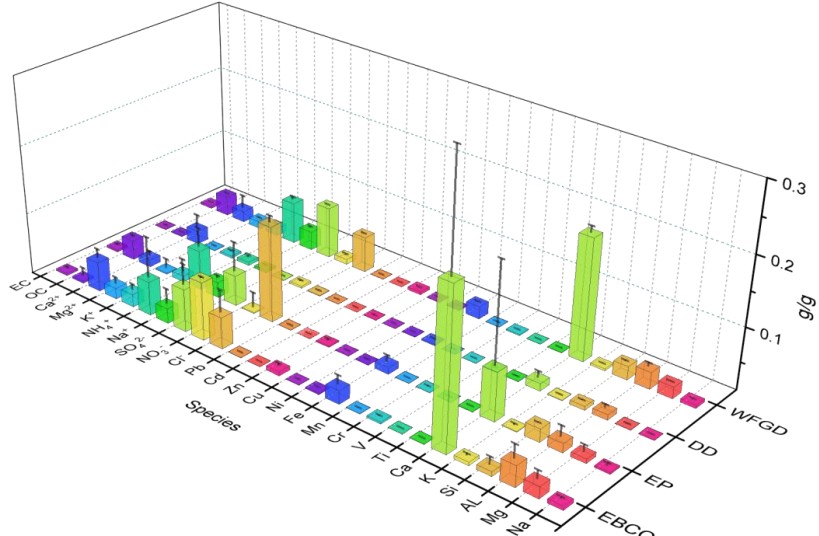


**Figure 4.** Comparison of $PM_{2.5}$ source profiles collected under different dust removal and
desulfurization facilities. EP denotes electrostatic precipitators, EBCC denotes electric bag
compound dust collectors, WFGD denotes wet flue gas desulfurization, DD denotes dry
desulfurization. Data from the source library of Nankai University) were counted.

To evaluate the impact of different sampling methods on the contents of source





profiles, measurements with the resuspension sampling method (RSM) and the
dilution tunnel sampling method (DTSM) were simultaneously used for source
sampling at a coal-fired power plant in Wuxi, China were compared. The results of
the obtained $PM_{10}$ source profiles are shown in Fig. 5. For RSM, the crustal elements
(Si) and carbon components (OC, EC) are significantly higher than DTSM. The $SO_4^{2-}$
content of DTSM is significantly higher than RSM, reaching 0.1600 g/g. And V, Cr,
Mn, Co, Ni, Cu, Zn, Pb and other trace metal fractions are strongly enriched in DTSM,
which is 1.4 to 100 times that in RSM, suggesting that these trace metal elements have
a low melting point and are easily liquefied or gasified during combustion, and then
condensed on the surface of the particles in the flue or after exiting the flue (where
small particles have a large specific surface area and are more prone to enrichment)
(Dai et al., 1987). The similar results were also reported earlier elsewhere (Meij,
1994;Meij and Winkel, 2004;Zhang et al., 2009b).

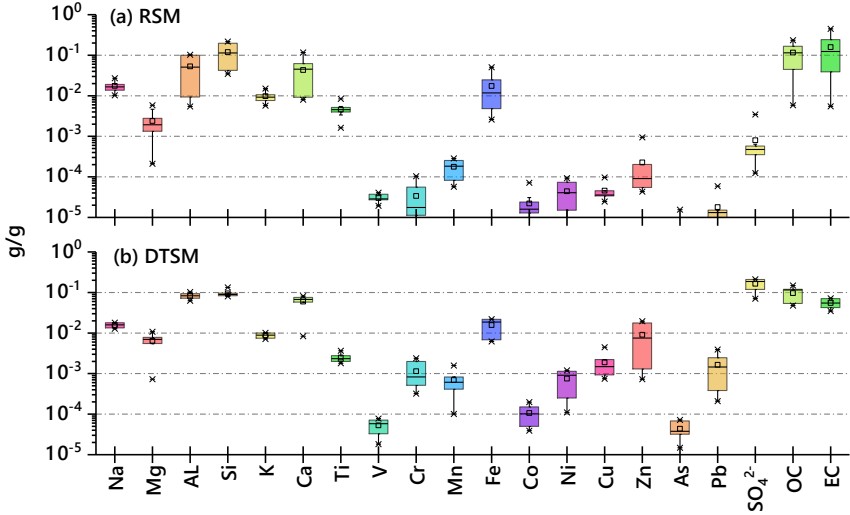




**Figure 5**. Characteristics of chemical profiles for PM$_{10}$ emitted from coal-fired power plant
obtained by different sampling methods in Wuxi city. RSM and DTSM denote resuspension
sampling method and the dilution tunnel sampling method, respectively. Data from the source
library of Nankai University were counted.

The PM speciation profiles of coal-fired sources have rarely been reported in China
(Kong et al., 2011). Comparing the main components of coal combustion PM$_{2.5}$ source
profiles derived from several other published source profiles (Chow et al., 2004;Liu et
al., 2016;Xia et al., 2017;U.S.EPA, 2014), the SO$_4^{2-}$, Ca$^{2+}$ and OC have significantly
higher abundances than other components in China and USA, but there are also large
variations in species abundances. The most diverse components are NO$_3^-$ and S. The
difference between these two components with less content may not only be related to
the coal nature and the dust removal equipment, but also the determination method of
the components and the sensitivity of the corresponding instrument (Xia et al., 2017).
As we mentioned above, there are many factors that affecting the profiles of coal
combustion sources. Therefore, when performing source apportionment study, local
source profiles have the priority to be measured in the study area. To improve the
accuracy and reliability of source apportionment results, it is necessary to measure the
local sources in real-world to avoid blindly drawing on the foreign source profiles.
The industrial emissions are one of the most important sources in China (Zhu et al.,
2018). Particles from industrial emissions is mainly collected by dilution tunnel
sampling method. The source profiles of industrial emissions could be influenced by





several key factors, such as raw materials used in industrial processes, manufacture
processes, various sampling methods, different sampling site, control measures taken
by different factories and process operating conditions (Watson and Chow, 2001;Kong
et al., 2011;Pant and Harrison, 2012;Guo et al., 2017). The primary source profiles of
industrial emissions in China include cement plant, steel plant and coking plant. Fig. 6
shows the chemical composition of China's main industrial emissions (cement plant,
coking plant and steel plant) (Ma et al., 2015;Qi et al., 2015;Yan et al., 2016;Zhao et
al., 2015a). There are great differences between the source profiles from different
industrial sources. For cement industrial sources, $Ca^{2+}$, Al, OC and $SO_4^{2-}$ are the most
abundant species, with average value less than 0.0010 g/g. For coking industrial
sources, $Ca^{2+}$, Al and $SO_4^{2-}$ are elevated while OC displayed a somewhat notable
lower level. For steel industrial sources, the highest fraction species are Fe, Si, K and
$SO_4^{2-}$ , while $Cl^-$, $Ca^{2+}$, EC and OC showed a lower content less than 0.0010 g/g.
In China, there are many industrial types with different emission characteristics. The
source profiles of industrial emissions are far from being fully understood so far. The
profiles of some important industrial sources, such as the glass melt kiln, non-ferrous
smelting, and ceramics, are still unknown and needed further investigation in the
future.



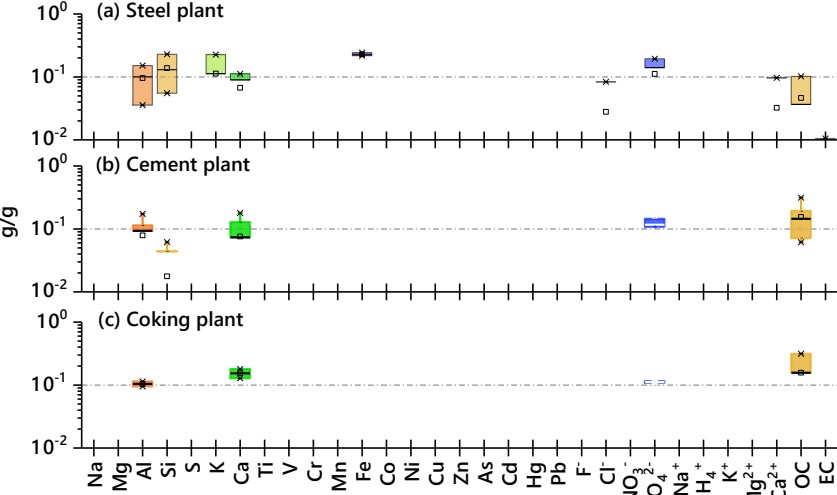


**Figure 6**. Characteristics of chemical profiles for particulate matter emitted from industrial


emissions. Data from the source library of Nankai University, Zhan et al. (2015), Qi et al.


(2015), Ma et al. (2015) and Yan et al. (2016) were counted.



**2.2.2 Vehicle emissions**


Vehicle emissions is appears to be the predominant source of ambient PM$_{2.5}$ in urban


areas in China (Cai et al., 2017b;Cui et al., 2016;Zhang et al., 2015). It is reported that


the contribution of vehicle emissions to PM$_{2.5}$ is in the range of 5% to 34% over China


based on receptor models (Zhang et al., 2017b). Given that there are many factors


affecting vehicle emissions such as fuel types, vehicle types, emission control


technologies, operating conditions, engine performance, sampling methods and so on


(Watson et al., 1990;Chen et al., 2017b;Maricq, 2007). The representative of the


source profiles of vehicle emissions are often controversial. Generally, there are two


methods for the sampling of vehicle emissions: direct sampling method (DSM) and






source dominated sampling method (SDSM) (e.g., measured in tunnel, on parking lot
and roadside.) (Kong and Bai, 2013). Fig. 7 summarizes the $PM_{10}$ source profiles of
different vehicle types obtained by direct sampling method in China (Chen et al.,
2017b). For both diesel and gasoline vehicles, their emission profiles are dominated
by OC, EC, $NO_3^-$, $NH_4^+$, $SO_4^{2-}$, Ca, Fe and Zn. While the abundance of EC in diesel
vehicle exhaust (particularly in heavy-duty diesel vehicle exhaust) is much higher
than that in gasoline vehicles, which may due to the different combustion completion
rates between diesel and gasoline on account of the length of hydrocarbon chains of
them (Chen et al., 2017b). Since Mn has been used in the gasoline explosion-proof
agent, the fraction of Mn in the particulate matter from the gasoline vehicle emission
is higher than that of the diesel vehicle.

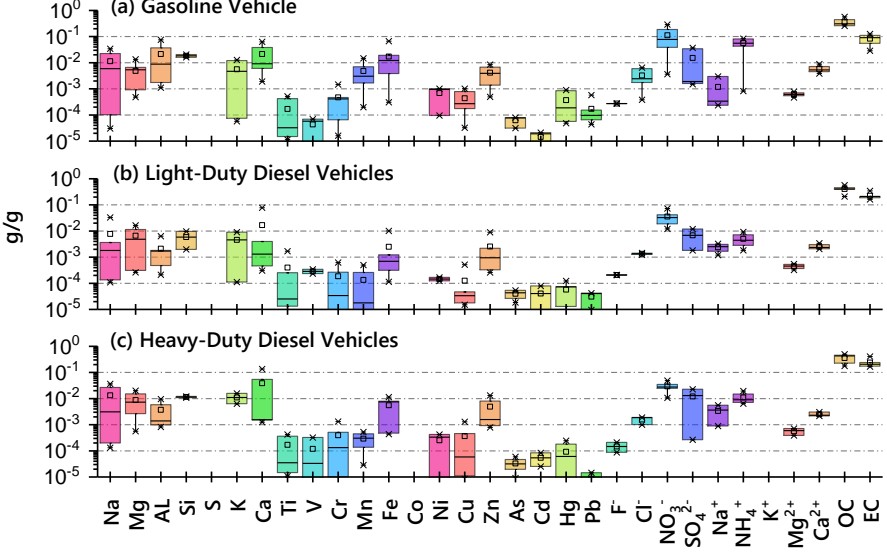


**Figure 7**. Chemical compositions of source profiles for $PM_{10}$ of different vehicle types

obtained by direct sampling method. Data from the source library of Nankai University and



Chen et al. (2017) were counted.

Fig. S1 summarizes the characteristics of chemical profiles for particulate matter
emitted from vehicles obtained by different sampling methods. Crustal elements (Si,
Al, Ca, Mn, etc.) in the chemical profiles obtained by SDSM are higher than that of
DSM, which may due to the influence of suspended road dust. $NH_4^+$ and $NO_3^-$ in
chemical profiles obtained by DSM are lower than that of SDSM, probably because
the volatile organic compounds and other precursors are still in the gaseous state when
the samples were collected at a higher temperature by DSM (Kong and Bai, 2013).
The source profiles of the vehicle exhaust alsovaried with fuel types, vehicle types
and vehicle age. In China, the oil used for vehicle has been upgraded for 5 times in the
past eighteen years. The evolutions of the fractions of Mn, Pb and $SO_4^{2-}$ in particulate
matter emitted by vehicle from the past three decades are shown in the Fig. 8. Pb was
a tracer of the gasoline before 2000 while leaded gasoline was banned to be used in
mainland China after 2000 (Zhang et al., 2009a). The standard value of sulfur in the
car-used gasoline is 800 µg/g in 2000 and 10 µg/g in 2018 (Guo, 2013). The standard
value of Mn is 0.018 g/L in 2000 and only 0.002 g/L in 2018 (Li, 2016). The similar
trend could also be found in the standard of diesel in China (Zhang et al., 2009a). All
these changes in the oil standard will definitely cause the evolution of source profiles
of vehicle exhaust. With the government's request to stop producing, selling and using
of leaded gasoline, the fraction of Pb in vehicle emissions decreased significantly. In
2005, the fraction of Pb in motor vehicle emissions dropped significantly as compared





with 1985 (Dai et al., 1986;Han et al., 2009). And the fraction of Mn is also greatly
reduced after 2000 (Bi et al., 2007;Han et al., 2009). After 2000, the fraction of $SO_4^{2-}$
in vehicle emissions also showed a significantly decreasing trend, indicating a causal
relationship with the reduction of sulfur in the car-used gasoline in China.

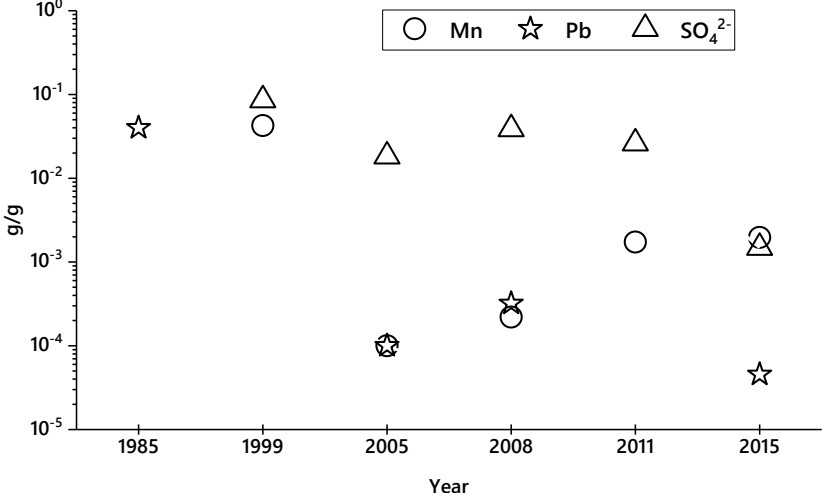


**Figure 8**. Time series of Mn, Pb and $SO_4^{2-}$ of the particulate matters emitted from vehicles
obtained. Data were collected from the source library of Nankai University, Zhang et al.
(2000), Guo et al. (2013), Li et al. (2016). Zhang et al. (2009), Dai et al. (1986), Han et al.
(2009) and Bi et al. (2007).
Xia et al. (2017) compared the main components of on-road vehicles $PM_{2.5}$ source
profiles derived from local studies and SPECIATE database, finding that both the
source profiles of motor vehicles in China and the United States were dominated by
OC and EC, but their proportions were quite different (Kong, 2012). In American, the
gasoline, ethanol and methanol are added as the aerator, while the oxygen content in





domestic gasoline is relatively small, which is an important reason for the difference
in the OC content in the spectrums at home and abroad (Xia et al., 2017). In China,
the fraction of $SO_4^{2-}$ is generally higher than that of foreign motor vehicles (Wang et
al., 2015;Xia et al., 2017), which may be related to the relatively backward
implementation of domestic oil standards and the high sulfur content (Guo, 2013;Li,

2016).


### 372    2.2.3 Fugitive dust

Fugitive dust is founded to be one of the major sources of urban particulate matter
(Chow et al., 2003;Kong et al., 2011;Cao et al., 2012;Zhu et al., 2018), especially in
northern cities in China with dry climate and limited precipitation (Shen et al.,
2016;Cao et al., 2008). Urban fugitive dust is not only influenced by soil properties
with geographic locations. In addition, it is actually the mixture of various
dust-related sources. Therefore, fugitive dust is often referred to soil dust, road dust,
construction dust et al (Doskey et al., 1999;Kong et al., 2014). Fugitive dust samples
were eventually collected by using resuspension chamber.
Fig. 9 shows that the primary species in soil dust are Si, Al, Ca, with mass fractions
ranged from 0.0500 to 0.2010 g/g. Si is the predominant species among the detected
elements, followed by Fe, Na and Mg. The main chemical components of road dust
are Si, OC and Ca, with fractions ranged from 0.0712 to 0.0855 g/g. Al, Fe and $SO_4^{2-}$
are the relatively lower species (less than 0.0005 g/g) in the chemical profiles of road
dust. Si, Ca, Al and Fe are all crustal elements, indicating that the soil dust has a



greater impact on the composition of road dust. It also shows that OC and $SO_4^{2-}$ in the
source profiles of road dust are higher than that of soil dust, indicating that the road
dust is also affected by vehicle emissions or coal combustion and other anthropogenic
sources (Ma et al., 2015). In general, the total water–soluble ions accounts for
0.0248-0.0648 g/g of fugitive dust, which suggests that insoluble matter is the main
component of fugitive dust.

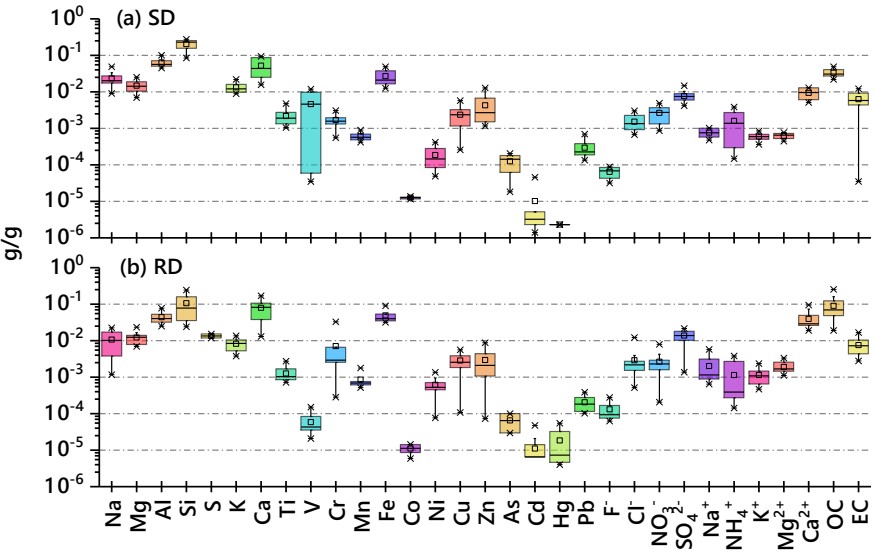


**Figure 9.** Characteristics of chemical profiles for particulate matter emitted from fugitive dust.
SD and RD denote soil dust and road dust, respectively. Data were collected from the source
library of Nankai University.

Many studies have demonstrated that ratios of different chemical components can be
used as markers for fugitive dust (Alfaro et al., 2003;Arimoto et al., 2004). Kong et al.





(2011) found that the Ca/Al ratio of paving road dust affected by construction
activities was significantly different from that of soil dust. Zhang et al. (2014)
reported that the heavy metals like Zn and Pb are able to be considered as the tracers
of urban fugitive dust, because they found Zn/Al and Pb/Al ratios in urban fugitive
dust were 1.5 to 5 times those in desert, Gobi, and loess soil samples. The $NO_3^-/SO_4^{2-}$
ratio has been used to compare the relative importance of stationary sources vs mobile
sources. Much high $NO_3^-/SO_4^{2-}$ ratio of road dust in Hong Kong has been reported by
Ho et al. (2003), revealing the more important impact of vehicle emissions on the
chemical composition of road dust as compared to coal combustion.

**2.2.4 Biomass burning**
Traditionally, China is an agricultural-based country in the world (Bi et al., 2007). As
an effective way to eliminate plant residues, direct combustion (open burning) is the
predominant and popular practice during the harvest seasons (Andreae and Merlet,
2001;Ni et al., 2017;Cheng et al., 2013;Li et al., 2014b;Streets et al., 2003), but it
releases a lot of pollutants into air, and consequently impacting air quality, health and
climate (Yao et al., 2017;Chen et al., 2017a). Biofuel burned with stoves is also an
important source of biomass burning (Tian et al., 2017). The wheat straw, corn stalks
and rice straw represent 80% of the agricultural combustion in China (Ni et al., 2017),
and there are also firewood, soybean and rape, etc. In addition to biofuel, sampling
procedures and conditions, there are great differences in the levels and chemical
properties of PM measured from different methods (Tian et al., 2017;Vicente and



Alves, 2018). At present, there are two popular ways in the measurements of biomass
burning: field combustion experiment (FCE) and laboratory combustion simulation
(LCS) (Hays et al., 2005;Li et al., 2014a;Sanchis et al., 2014;De Zarate et al., 2000).
Compared with other sources, reports on the profiles of the biomass burning in China
were rarely published. Fig. 10 summarizes the source profiles for $PM_{2.5}$ obtained by
different sampling method in China. The profiles of biomass boiler exhaust are
obtained by resuspension sampling method. The main components in the profiles of
biomass burning are OC, EC, $K^+$, $Cl^-$, K and Ca (Fig. 10). The fraction of EC is much
higher in the biomass boiler exhaust than the laboratory combustion simulation,
showing that the limitation and the uneven mixing of the air in the biomass boiler is
easy to cause straw to burn in anaerobic condition (Tian et al., 2017), and make the
emission of EC higher. The oxygen content is relatively sufficient in open-air
combustion, which leads to relatively high OC emission. The fraction of Ca was much
higher in biomass boiler exhaust than in field measurements (Fig. 10).
Field combustion experiment (FCE) is closer to the actual conditions of outdoor
combustion of straw, but the experimental condition is difficult to control well. In
consideration of the relatively small burning amount of straw in the LCS and a certain
difference with the actual environment conditions of the FCE, the LCS can better
control the combustion conditions (Wang et al., 2016). Due to    the    different
temperature between FCE and LCS (Jensen et al., 2000), a clear different release of
$K^+$ and $Cl^-$ in $PM_{2.5}$ emissions to atmosphere (Fig. 9).
For specific components of the emissions from the biomass burning, EC emissions



from firewood combustion was highest, which is mainly due to the higher content of
lignin in wood (Tang et al., 2014). The content of lignin make for the formation of
black carbon (Wiinikka and Gebart, 2005). At the same time, the content of volatile
components of the firewood is relatively high, and the structure is dense, making it
easy to combust completely in the furnace, reducing the production of OC.

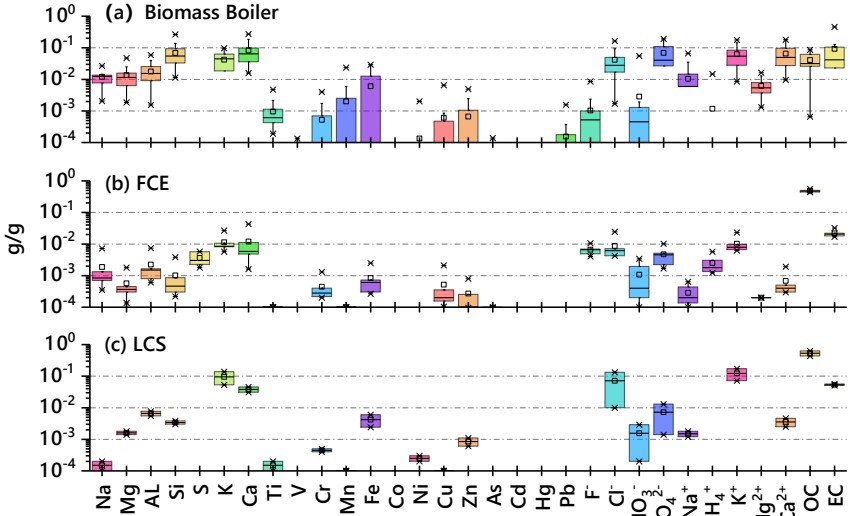


**Figure 10**. Major chemical compositions of₅ source profiles of biomass burning PM₂obtained

by different sampling method. FCE and LCS denote field combustion experiment and
laboratory combustion simulation, respectively. Data were collected from the source library of
Nankai University.

Chen et al. (2007) investigated the particulate emissions from wildland fuels burning
in a laboratory combustion facility in the U.S., and found the percentage of TC of PM
was 63.7% ~ 100%, which was higher than that in China (4.9%~68%). In addition,



adsorbed organic vapors were detected as OC in the experiment conducted by Chen et
al. (2007), resulting in increased content of OC and TC. K (0.4%~23.7%), Cl
(0.1%~9.6%) and S (0.1%~2.9%) were important part of the remaining PM mass in
the U.S, which is different from China due to the different biomass categories and
combustion processes.

**2.2.5 Cooking emissions**
With the economic growing, the types of food ingredients on the table and cooking
styles have gradually become more diverse. Since 1990s, the variety of ingredients
and cooking styles was also influenced by the foreign food culture. As China is
famous for its food culture, various cooking styles can be found in different regions,
even in different cities. Thus, cooking is undoubtedly an important local source of
ambient particles. Given that there is no ubiquitous source profile for cooking
emission, it is better to measure source profile of cooking emissions in real-world in
the study area. As one of the essential cooking ingredients in the food and beverage
industry, the types of edible oils are changing in recent years (Pei et al., 2016).
Soybean oil, rapeseed oil and peanut oil are common edible oils for public dining.
Due to changes in consumer demand, other types of edible oils, such as olive oil,
camellia oil and flaxseed oil, have also been increasingly welcomed by the catering
industry. Furthermore, Chinese-style cooking is characterized by high temperature
stir-frying that releasing much more organic matter than the cooking style of western
food (Zhao et al., 2007b).





The chemical nature of $PM_{2.5}$ emitted from commercial cooking were investigated in
many studies, with source profiles varied greatly with different factors such as
cooking styles, cooking foods, seed oils, fuel, et al (He et al., 2004;Zhao et al.,
2007b;Hou et al., 2008b;Zhao et al., 2015b;Pei et al., 2016). Robinson et al. (2006)
found that the contribution of cooking emission to OC in $PM_{2.5}$ calculated by
chemical mass balance model using different source profiles yielded a difference by a
factor of more than 9.
Studies founded that organic matter accounted for 66.9 % of the TSP mass emitted
from cooking activities (Zhao et al., 2015b). OC is the major constituent and
accounted for 36.2%~42.9% of the total mass, while the fraction of EC is much lower.
Several water-soluble ions measured in the fine particles emission presented a
relatively lower but a noticeable percentages, which made up of about 9.1%~17.5% of
the total $PM_{2.5}$ mass (Anwar et al., 2004). Inorganic elements are found to contribute
about 7.3%~12.0% of the total $PM_{2.5}$ mass due to their greater presence in cooking oil
and raw materials (He et al., 2004).
Fig. 11 shows the $PM_{2.5}$ chemical profiles of cooking emissions including hot pot,
Chinese restaurant, barbecue and cafeteria (See and Balasubramanian, 2006;Taner et
al., 2013;Zhang et al., 2017a). For elements, on average, the most abundant elements
in cooking profiles is Al, followed by Ca and Fe. Similar results have also been
reported elsewhere. The high levels of Ca and Fe are probably emitted from raw
material and cooking utensils (See and Balasubramanian, 2006;Taner et al., 2013).
And the high level of Cr, originated from stainless steel grills, was observed in a





barbeque restaurant (Taner et al., 2013). Overall, OC is the most abundant species in
the profiles of cooking emissions.

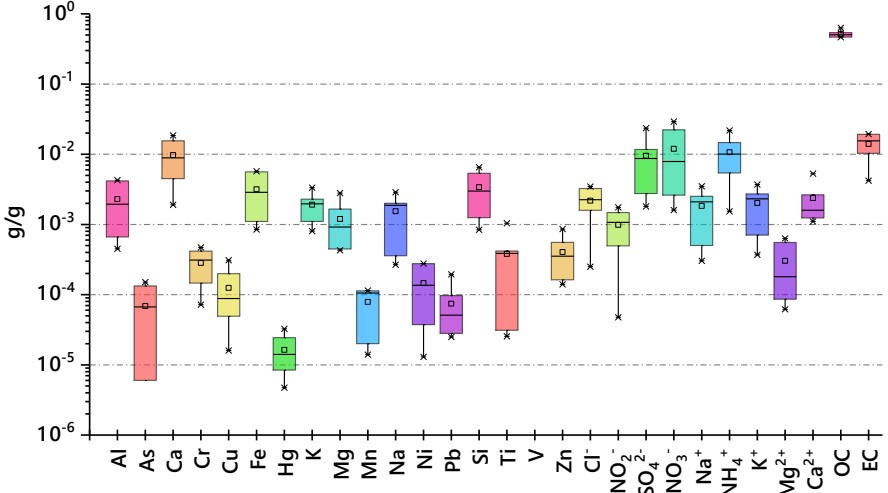


**Figure 11.** PM$_{2.5}$ Chemical profiles of cooking emissions. Data from the source library of

Nankai University, Zhang et al. (2017), See et al. (2006) and Taner et al. (2013) were counted.

Organic matter (OM) is the predominant species in PM$_{2.5}$ emitted from cooking
activities (He et al., 2004;Hou et al., 2008a;Pei et al., 2016).Many organic compounds,
including n-alkanes, dicarboxylic acids, polycyclic aromatic hydrocarbons (PAHs),
saturated fatty acids and unsaturated fatty acids, were quantified in the above studies.
Fig. 12 shows the fractions of main organic compounds in the quantified OM
emission from residential cooking (Zhao et al., 2015b) and commercial cooking (Pei
et al., 2016). Among the quantified organic compounds, the predominant species is
unsaturated fatty acids (49.4%-77.8%), followed by saturated fatty acids



(25.1%-43.8%).


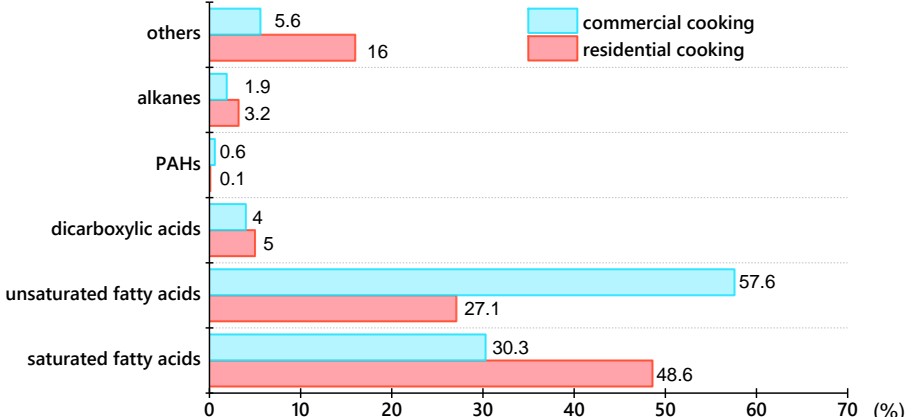


**Figure 12**. Proportions of major organic compounds in quantified OM emission from
commercial cooking (Pei et al., 2016) and residential cooking (Zhao et al., 2015b) .

In addition, except for biomass burning, many studies have reported that the
levoglucosan was also founded in the emissions from residential coal combustion
(Yan et al., 2017) and a variety of Chinese and western cooking styles (He et al.,
2004;Zhao et al., 2007b, a). Pei et al. (2016) also found Italian cooking style released
the smallest amount of monosaccharide anhydrides and the largest amount of
cholesterol due to the lower ratio of vegetables to meat used in the Italian cooking
than Chinese cooking materials. Malay cooking released higher PAHs concentrations
than the Chinese and India methods (See et al., 2006). Deep frying emitted more
PAHs than other cooking methods because of the higher temperature and more oil



used during cooking. As far as we know now, molecular markers used for cooking
included levoglucosan, galactosan and cholesterol (He et al., 2004;Zhao et al., 2007b,
a) while cholesterol can be regarded as a best marker for meat cooking (Schauer et al.,
1999;Schauer and Cass, 2000;Schauer et al., 2002).

**2.3  Source categories cluster analysis**
The chemical profile of a given source category was always established from profiles
of several similar sources belonging to this category. Non-negligible uncertainties
would be introduced in this process. To evaluate such uncertainties, the coefficient of
variation (CV, the standard deviation divided by the mean) is used in this section to
further characterize the homogeneity of sources within the same source category (Fig.

13).

The values of CV above 3 (Pernigotti et al., 2016) are observed in coal combustion,
industry emissions and biomass burning, indicating these source profiles shows a
great variation due to the large variations of their influencing factors as described in
above sections. The profiles of road dust and soil dust showed a less variation with the
stable chemical characteristics among the different profiles in the same category.
However, the response of source profiles to various impact factors is different (Fig.
13(a)-(c)). For example, the difference of coal combustion source profiles obtained by
resuspension sampling is greater than that by dilution tunnel sampling, while the
desulfurization method has less influence. For biomass burning, small differences
exist in the source profile established through FCE and LCS methods while quite



difference from that from biomass boilers. More details need further investigation.
Since source profiles owned local characteristic, it is important and necessary to
establish and update local source profiles to reveal the real situation of source
emissions (Zhang et al., 2017b;Zhu et al., 2018). However, local source profiles are
not always available in some developing areas in the case of limited founds and poor
instruments. According to the above statistical results, it can be inferred that the
profiles of road dust and soil dust could be references for the cities in China without
such local profiles, while it is necessary to establish the local profiles of the local
industrial emissions, vehicle emissions, coal combustion, and biomass burning, etc.

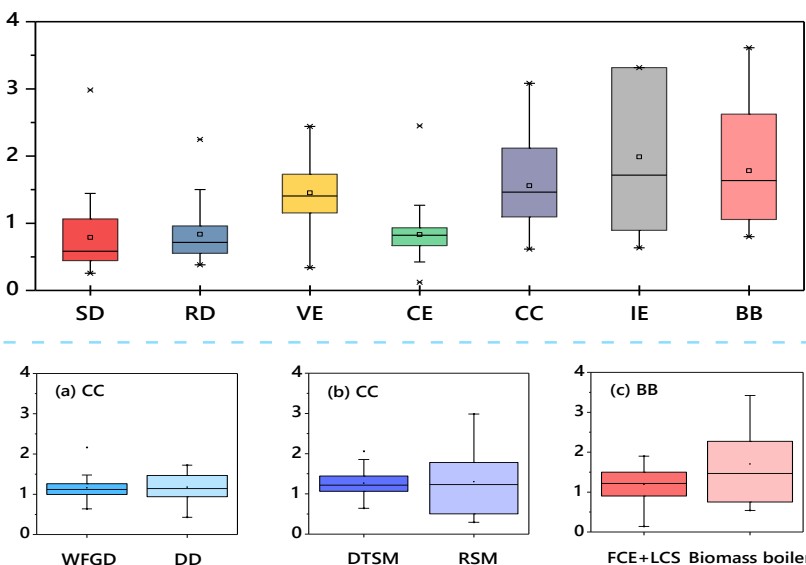

**Figure 13.** Coefficients of variation calculated for each source category. SD denotes soil dust,
RD denotes road dust, VE denotes vehicle emissions, CE denotes cooking emissions, CC
denotes coal combustion, IE denotes industrial emissions, BB denotes biomass burning,





WFGD denotes wet flue gas desulfurization, DD denotes dry desulfurization, DTSM denotes
the dilution tunnel sampling method, RSM denotes resuspension sampling method, FCE
denotes field combustion experiment, LCS denotes laboratory combustion simulation.

In order to attribute the real-world measured source profiles with homogeneous
chemical signature, cluster analysis was applied to the collected data by using the
package R pvclust (Suzuki and Shimodaira, 2006;Pernigotti et al., 2016). The
significance test was performed with resampling the data via bootstrap method. This
function is expected to assign to each cluster an approximated unbiased (AU) p-value
by hierarchic clustering (Shimodaira, 2002). More details about the operation steps of
this method are discussed earlier by Pernigotti et al. (2016). Moreover, the source
profiles involved in the cluster calculation must contain more than two common
species. In order to reduce the interference of different particle sizes, we used 214
source profiles of $PM_{2.5}$ for the calculation. The result of cluster analysis and
additional information of the source profiles are shown in Fig. 14 and Table S1. As
shown in Fig. 14, clusters are marked if the AU p-value $\geq$ 90 (values were reported in
red). It shows that the source profiles are divided into (1) biomass burning, (2)
cooking emissions, (3) vehicle emissions, (4) and (5) coal combustion, (6) soil dust, (7)
road dust, (8) industrial emissions. The result indicates that most of the measured
source profiles in China have their own characteristics, however, there are some
different sources mixed up (Fig. 14), indicating that the information of routine
measured components such as elements, ions and carbon fractions in these profiles is



probably not enough to distinguish all the source categories. Both the source profiles
of cooking and vehicle emissions are characterized by high OC, which makes them
easy to be identified as the same source type. And the chemical collinearity of the
source composition between coal combustion and dust also makes it difficult to be
distinguished. To solve the chemical co-linearity problem between sources, more
specific tracers, especially organics should be further explored.

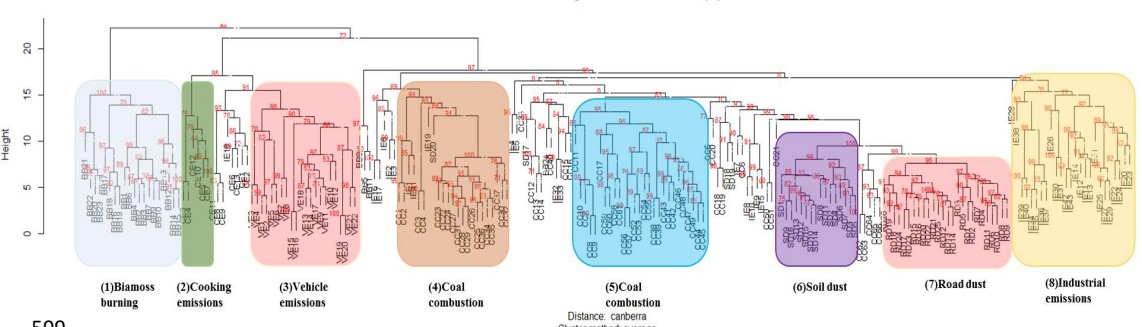


**Figure 14.** Result of cluster analysis on the profiles. AU p-values are reported in red
as %.

**3.  Conclusion**
The chemical profiles of main sources of particulate matter have been established in
China since 1980s. With the development of sampling and analysis techniques, the
dataset of source profiles have been gradually enlarged and could to able to reflect the
real emissions of the sources to the ambient air. A total of 374 published source
profiles, coupled with the database of source profiles (2870 profiles) founded by
Nankai University are reviewed in this work. Six source categories include coal



combustion, industrial emissions, vehicle emissions, fugitive dust, biomass burning
and cooking emissions are investigated to characterize sources in chemical nature and
explore the main factors that influencing the chemical composition. This effort gives
insights into the development of source profiles in terms of its applications in receptor
models, air quality models, validation of emission inventories and estimation of
source-specific emissions of individual compounds.
In general, coal combustion is the most complicated source in all source categories as
it is influenced by many factors from the fuel combustion processes to
pollution-controlling processes. Sulfate is the predominant species emission from coal
combustion source equipped with wet flue gas desulfurization device. The source
profiles of industrial emissions are mainly determined by the components of the
industrial products and its pollution-controlling techniques. With the changing
standards of gasoline and diesel oil since 1980, Pb and Mn are no longer the tracers of
emission from the gasoline vehicles. OC and EC are always the dominant species of
vehicle emissions from 1980s despite the changing standards. The profiles of the
fugitive dust including the road dust and soil dust are characterized by the high levels
of crustal elements, such as Si, Al and Ca. The profiles of the biomass burning are
determined by the biomass categories and the different combustion phases
(smoldering and flaming), with $K^+$ and levoglucosan to be the tracers. As for cooking
emissions, the source profiles of the emissions from the different cooking types were
all dominated by OC.
The uncertainty analysis of all these source profiles are undertook to explore the





variations of the different source profiles in the same source category and evaluate the
differences between source categories. A relatively large variation has been founded in
the source profiles of industry emissions, vehicle emissions, coal combustion and
biomass burning, indicating that it is necessary to establish the local profiles for these
source due to their high uncertainties. While the profiles of road dust and soil dust
present a less variation with the stable chemical characteristics among the different
profiles in the same category, suggesting that the profiles of these sources could be
referenced for the cities in China when the local profiles are not available. Since
source profiles owned local characteristic, it is important and necessary to establish
and update local source profiles to reveal the real situation of source emissions.
The result of cluster analysis on the routine measured species of source profiles
suggested that industrial emissions are quite homogeneous, but some of the sources
are difficult to be distinguished (cooking emissions vs vehicle emissions), indicating
that more chemical tracers, such as the isotopes and organic compounds, should be
further explored in the source profiles to reduce the collinearity among different
source profiles. In addition to chemical components, physical information (for
example, size distribution), is also an important property of particles that has a much
higher potential to be used to discriminate sources. There are hundreds of sources of
particulate matter in the real world, however, current database of source profiles still
lacking some important source categories that have significant impacts on the air
quality, especially the industrial emissions, such as the glass melt kiln, nonferrous
metal smelting, bricks and tiles kiln etc. Thus, specific focus should be placed on



these important but overlooked sources and the source profiles should be focused on
the sub-type in the future.

**Acknowledgements**
This work was financially supported by the National Key R&D Program of China
(Grant No. 2016YFC0208500 (No. 2016YFC0208501)) and the Fundamental
Research Funds for the Central Universities of China. The authors thank Jing Ding,
Xian Ma, Jiamei Yang, Tingkun Li, Jinsheng Zhang, Xin Du, Baoshuang Liu, Ming
Zhou and other students in our research group for their assistances in the literature
research, source sampling and post-chemical analysis related to this work.

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
