# Peer review of "Characteristics of the main primary source profiles of particulate matter"

_Atmospheric Chemistry and Physics, 2018_

## Referee Comment (RC1) · Anonymous Referee #2 · 23 Sep 2018

This study reviewed particle chemical composition profiles from typical emission sources measured in China, based on the 374 published profiles in literature and 2870 profiled conducted by the research team. Source profile is vital in source apportionment and pollution controls, and localized source information is important for accurate source identification and contribution estimation. The review made a significant contribution to this important area. But the manuscript in its present form needs clear clarification and revisions on its data analysis and presentation. My comments on this manuscript are as follows:

About literature search and paper selection, as done in many review studies, more de-

[Figure]

tails on the literature search and selection should be provided and evaluated. For example, how many people did the search and were they done independently? Are there any duplicated papers from different searchers and different database? How many papers found in total in the first round of the search? What're the inclusion/exclusion criteria? Was there any evaluation on the quality control and assurance in the decision of inclusion/exclusion? Line 147 and figure 2, - clarification of the sampling method into these three groups is inappropriate. e.g. "medium-volume" could be used in "dilution sampling" -they considered different factors instead of different approaches of the same factor. The authors should rethink of the classification and reanalyze the part. A similar problem of classification is in the biomass burning part- Biomass boiler, FCE (field combustion experiment), LCS (laboratory chamber study), the first refers to the burning facility, while the other two are experiment methods. How could these three be in parallel here? The study analyzed sources in "coal combustion, industrial emissions, vehicle emissions, dust, cooking emissions, biomass burning" – some are based on fuel type, some are sectoral difference- this mixed classification should be corrected. e.g. "coal combustion and industrial emissions"- is coal burned in many industrial factories? What's the burning fuel in industrial combustion process? Residential burning is a significant part of coal combustion, particularly in China. But the study did not have analysis and discussion on this. This important source should not be missed. Line 166, was TOT method used in EC/OC analysis? Line 180, retene is also one widely used tracers for biomass burning, especially wood combustion. Line 263, as seen in Figure 1 there is a significant number of coal emission studies, and even higher than the number of vehicle emissions. "rarely" might be inappropriate here. Line 425- may be good to provide some local studies on biomass burning emissions from indoor stoves and open burning. Line 427 and figure 10- as mentioned above, "biomass boiler" was not a "sampling method". because of the problem in such classification, the discussion from line 430 needs to be revised. Line 428, is "biomass boiler" the boilers used in industry or household? What about the home stoves? Line 433-434- high EC or BC emissions are usually due to the domination of high temperature flaming burning conditions. Lien

437-438, it seems that the authors only considered open burning when referring to "FCE". Lines 441-443, a recent study showed that for open biomass burning, emission factors of most air pollutants from field measurements and simulated chamber studies in laboratory are comparable. Line 446, also due to high combustion temperature and flaming dominance burning conditions. Line 527, the reference "Yan et al., 2017" is missing Line 555-556, as mentioned above the biomass boilers are burning facilities while the other two are sampling/experiment methods. Line 600, please consider enlarging and improving the resolution Giving different burning conditions and different fuel properties even for the same type of fuel, within the current compiled database, it is interesting to look into the spatial difference in source profiles for the same source type. Generally, the manuscript is understandable but could be improved after a language edit and polish. Please go through the manuscript and check for grammars and spelling errors for example, Line 79, "source measurement is actually it is time to . . ." – the sentence is incomplete. Please check and revise

---

## Referee Comment (RC2) · Anonymous Referee #4 · 2 Oct 2018

This manuscript presented detailed characteristics of the main primary source profiles of PM in China. The conclusions here can provide clear evidences for the source apportionment and environmental management. Reviews of evolutions of sampling methods, chemical analytical methods and source profiles were also given. Besides, the authors also proposed future requirements for the development of source profiles in China. However, some descriptions in the manuscript need to be further improved, and more tracer characteristics of the sources need to be discussed. This manuscript can be considered for publication in Atmospheric Chemistry and Physics after major revision. Specific Comments: 1. Line 79-80, rewrite the sentence. 2. Line 96-97, except the key words listed in the manuscript, have you ever considered other key

words, such as coal combustion, industrial emissions... for the literature research? 3. Section 2.2.1, it was shown in Figure 1 that residential coal combustion contributed 20 literatures, but it has been completely neglected in this section. So far as I know, PM emitted from residential coal combustion is quite different from IBW and PPW. Please give further discussion about residential coal combustion. 4. Figure 4, Only OC, EC and Ca were described in line 221-225, however, other components such as $NO_3-$, Cl-, and $Ca2+$ also varied significantly between EP and EBCC, please give more descriptions; Only $SO_4^{2-}$, Ca and OC were described in line 226-234, how about other components such as $NH_4+$, Na+ and Cl-? Ca and $Ca2+$ showed opposite tendencies between DD and WFGD, please give reasonable explanation. 5. Line 246-247, it is said that Si, OC and EC from RSM are significantly higher than DTSM, however, it is showed from Figure 5 that Si and OC had almost the same medium value and average value for RSM and DTSM, I do not agree about this conclusion. 6. Line 263, a total of 71 literatures are showed in Figure 1, why said "rarely" here? 7. Mn and Pb in Figure 7 showed characteristics only can correspondence with gasoline vehicles in 2015 (Figure 8). So, are the data in Figure 7 and Figure 8 from the same data source? And which year? Can they represent the vehicle emissions? Are the vehicles in Figure 8 gasoline vehicles? 8. Line 337, a space is missed between "also" and "varied". 9. Line 396, full names of SD and RD should be given for the first appearance. 10. Line 441-443, rewrite the sentence. 11. Line 434, it was mentioned that the relatively sufficient oxygen content could help for the OC formation, and in Line 449, the complete combustion was considered can reduce the production of OC, please give reasonable explanation. 12. Line 452, check the spellings. 13. Section 2.2, characteristics of PM from several sources were discussed here, however, in my opinion, it is better to give more tracers or distinguished features of each source, which can make it easier to identify different sources.

Please also note the supplement to this comment:
https://www.atmos-chem-phys-discuss.net/acp-2018-687/acp-2018-687-RC2-

supplement.pdf

---

## Referee Comment (RC3) · Anonymous Referee #3 · 2 Oct 2018

The authors investigated the evolution of primary source profiles of PM in China between 1987-2017. They reviewed a total of 3244 chemical profiles, assessed their uncertainties, and conducted a cluster analysis to analyze the heterogeneity across different source categories. There are many studies in literature that have summarized the characteristics of PM source profiles in China. Compared to the previous studies, the method used here is not novel, and I don't see much scientific significance in this paper though it summarized plenty of data and did some analysis. The paper is not well written and needs lots of editing. My major comments are as follows.

(1) After reading the title, I expected the evolution of source profile with time was one

[Figure]

research focus. However, the paper only analyzed the evolution of source profiles from vehicle emissions. I suggest the authors provide more discussions on other important sources (e.g., coal combustion and industrial emissions) if possible.

(2) Abstract. The authors mentioned "the most complicated profiles are likely attributed to coal combustion and industrial emissions."(Line 17). This is well recognized thus not appropriate to repeat it in the abstract. Please focus on the main findings of this study. For example, the results of cluster analysis should be summarized in the abstract.

(3) Introduction. The introduction part presents weak literature reviews. A literature review is much more than a descriptive list of materials available.

(4) Method. It is not clear to me how the authors selected the source profile that is of acceptable quality. What is the criteria for inclusion or exclusion of a profile from a literature? It is important that the method part is self-contained and clear enough for audiences to reproduce the given results.

(5) Section 2.3. The title need to be reconsidered since this section contains the analysis using the coefficient of variation as well.

(6) As a significant source, residential coal combustion is missed in the paper. Please provide more discussions.

(7) Line 184, the description of VOCs source profiles seems not quite related to the topic of this paper.

(8) Line 246 and figure 5, please check the figure and raw data if Si and carbon components for RSM are significantly higher than DTSM.

(9) Figure 11, please clarify the information of the chemical profile given here, i.e., is it an average profile or related to a specific cooking style?

(10) Many syntax and spelling errors in the text. For example, Line 33, "While the profiles of road dust and soil dust. . . . . .."; Line 307, "Given that there are many

factors. . .. . .".

---

## Referee Comment (RC4) · Anonymous Referee #4 · 3 Oct 2018

The knowledge of source profiles in China is significantly inadequate. In this manuscript, the authors aimed to reviewed the characteristics and evolution of source profiles in China from 1987 to 2017, which would provide very necessary information for source apportionment and evaluation of health effect from different sources. But, ACP as one of the high level paper at area of atmosphere research, the manuscript should be revised largely to deep discussed the evolution of source profiles. The latest version was considered without compact structure and profound discussion. I would like to review again after some major revision done.

Major:

1. Although it was reported that 3244 chemical profiles was discussed in this study, the authors should consider how could those database of profiles be used by other researchers? The latest version couldn't show the huge amount of data. It seems that some table for profiles were better than figure.

2. The structure of manuscript was not compact, etc. part 2.1. The manuscript should be written more logic.

3. One of the most important aims of this manuscript is to evolution the changes of profiles from 1987 to 2017. However, some profiles like coal combustion couldn't shown this trend. It should be better discussion from some aspects like source profiles variation from different years, processing and sampling methods.

4. It seems that some source profiles in China were not included in this review. I suggested that the authors should search more carefully. For example, the amount of source profiles for diesel emission published already would never be so small.

5. Many sentences were long and complicated, which were hard to understand. Some short and simple sentence should be better (etc. lines50-53; 59-63; 79-80; 124-126…).

6. It would be better that give some review about source profiles with organic matter, isotopes and size distribution (according to line 71-75).

Minor:

1. Line 65-66: add more typical research about source profiles. Line 71-73: add references.

2. Line 96-100: changed the sentence into passive "…were used as the key words…", and delete "searching for papers and dissertations".

3. Line 100: delete "the source profile data were compiled".

4. There is not shown the size distribution in Figure 1 (lines 106-108).

5. How about the source profiles detected in different areas?(part 2) (give the data marked in map is better)

6. Line 120: is it source profile research not source apportionment research? What the meaning of catch?

7. Variations of sampling methods during different periods were more important (Figure 2).

8. Line 181: check the format of comma.

9. Check the format of citation all of the manuscript.

10. Line 333-336: I wondered that the precursors of NO32- and NH4+ were VOC?
11. Check line 337, lines 339-340. Some sentence seems were copy by other places, which color was different with the normal.
12. Figure 8: please explain why the trend of Mn was increasing after 2005?
13. It would be better that some tables or figure to comparing the difference of source profiles between China and EPA (lines 360-370).
14. Figure 14 was hard to read.
15. It would be better to give the fractions of typical species to PM for each profiles, which could be evaluated whether the dominant species could be used as biomarker.
16. Please rewrite the conclusion.

---

## Referee Comment (RC5) · Anonymous Referee #1 · 13 Oct 2018

(1) The introduction should be improved, to give more description of source profiles and its importance. Also, as a review paper, the developing history and shortages for current source profiles should be better summarized. The science implication should be highlighted. (2) As the introduction of a review articles, all related references should be added. For example, Line 72-75, references for organic compounds, isotope and size distribution should be all listed, not just listing some examples. (3) The word evolution may be not suitable for the review of source profile. I believe change or variation is more suitable. (4) The authors just use the source profile related keywords which may miss some important papers. For example, you could not fine these key words in some tunnel or engine test studies. Also, the Elsevier database is not enough.

Such as papers published on the journals of ACS, AGU, Springer will be missed. (5) In the discussion section, more discussion should be added, not just say the higher or lower of components. Why they are higher or lower? For example, line 210-211 (6) Line 131-132, the sentence indicated dilution sampling has been widely used, but the author just listed one paper. Li et al., 2009 is only for household biofuel burning test. There are many sentences have the same problem. That is, the author just listed one paper to say something. It is not suitable, especially for review articles. Such as Line 142-143, Line 179-183, Line 191-194 (7) In figure 2, change the medium-volume sampling, there are also low-volume sampling methods used in source profile researches. Also in this figure, the sampling methods for vehicle emission should be given. (8) Line 180, what is azzaarenes? It is a component or a type of components? Also the author use "a marker" which is false for plural. Same problem in Line 182. (9) Line 181, the references should be cited by year. The dot "ïïjŇ" should be in English "," (10) All the description about VOCs should be deleted in the paper. (11) Line 232-233, why wet desulfurization can cause the conversion of organics to OC? (12) In the discussion part, some sentences are not quantitatively. For example, Line 448-449, the content of volatile components of the firewood is relatively high. The authors should collected the data for volatile materials for different types of fuels and give more reliable results. Line 431-433, "much higher" indicated how much higher? (13) Line 390-391, how can the water-soluble ions contents itself suggests that insoluble matter is the main component? For many soluble components, the previous studies may not analyze them. The authors can only conclude which component are more soluble, but not for particles. (14) Line 381-383, the author say Si is the predominant species, please give the mass percentages of Si in all the elements, not its content level. Similar description in other places. (15) Line 367, Line 365, "generally higher", "relatively small", please give data; (16) Line 363, their proportions were quite different, please give data; (17) Line 351, I think it should be after 2011. Also, for the profiles, how can the authors know the source samples were just collected in 2005, 2008, and so on? Maybe the research published in 2011, but the samples were for older cars than

2008 or even 2005. (18) Line 442, different temperature between FCE and LCS, you mean the burning temperature or the sampling temperature. For the sampling test in LCS, dilution tunnel always reduced the high temperature flume gases to ambient temperature. I guess, it should be the Cl- depletion for ambient field sampling. The English should be improved and there are also obvious errors. I can just list some: (1) Line 79, the sentence should be corrected; (2) Line 304, "is" into "are"

---

## Author Comment (AC1) · 5 Dec 2018

Response to comments:

We sincerely thank the reviewer for his/her helpful comments and guidance. Addressing the major points raised during the review process has substantially improved the quality of the manuscript. We have provided responses to each reviewer comment below in blue.

**Reviewer #2:**

This study reviewed particle chemical composition profiles from typical emission sources measured in China, based on the 374 published profiles in literature and 2870 profiles conducted by the research team. Source profile is vital in source apportionment and pollution controls, and localized source information is important for accurate source identification and contribution estimation. The review made a significant contribution to this important area. But the manuscript in its present form needs clear clarification and revisions on its data analysis and presentation. My comments on this manuscript are as follows:

Major comment 1:

About literature search and paper selection, as done in many review studies, more details on the literature search and selection should be provided and evaluated. For example, how many people did the search and were they done independently? Are there any duplicated papers from different searchers and different database? How many papers found in total in the first round of the search? What's the inclusion/exclusion criteria? Was there any evaluation on the quality control and assurance in the decision of inclusion/exclusion?

Response:

This is a great point. We now have searched the literatures again based on a two-round paper search work and using more source-related key words.

Author's changes in manuscript:

Our revision to this section is included in the following bulleted list:

1. Addition to the Introduction section.

"The main ubiquitous sources of atmospheric PM in China during the past three decades can be roughly divided into coal combustion sources (CC, with sub-type sources of

coal-fired power plants, coal-fired boiler from industry and residential coal combustion), vehicle exhaust (VE, gasoline and diesel engines), industrial processes emissions (IE), biomass burning (BB), cooking emissions (CE), fugitive dust (FD, with sub-type sources of soil fugitive dust, construction dust and road dust) and other localized specific sources."

2. Details on the literature search of above sources has been added to the Introduction section in response to this comment.

"To collect the potential published data related to source profiles, a two-round literature search work covering literature from 1980-2018 was done in this work. In the first round of searching, two authors are responsible for the same source to ensure every source category has been searched twice independently. The search keywords depend on source category. The following keywords for each source were used individually or in combination. As for CC sources, the key words are "coal combustion/coal burning/coal-fired boiler/coal-fired power plant/residential coal" and "source profile/chemical profile/particle composition". The key words for other sources are shown as follows. IE: "industrial emission" and "source profile/chemical profile/particle composition"; VE: "vehicle emission/exhaust emission/traffic emission/diesel engine/truck emission/gasoline engine/on-road vehicle/tunnel experiment/chassis dynamometer/portable emission measurement system" and "source profile/chemical profile/particle composition"; CE: "cooking emission" and "source profile/chemical profile/particle composition"; BB: "biomass burning/bio-fuel boiler" and "source profile/chemical profile/particle composition"; FD: "soil/fugitive dust/crustal material/construction dust/road dust" and "source profile/chemical profile/particle composition". Papers and dissertations in Chinese on China National Knowledge Infrastructure (CNKI) and papers in English on the web of science were searched using above keywords, respectively. The duplicated paper was then double-checked and excluded. The papers with topic related to source profiles but without providing any information of real-measured sources were also excluded. For example, papers reported source apportionment results with the use of PMF and CMB but without

reporting local profiles were not taken into account. As a result, a total of 193 papers have been collected from these efforts. In the second round of searching, the valid papers with available source profile data and detailed source sampling and chemical analysis methods were counted and used for post-analysis. Finally, a total of 456 published source profiles since the 1980s across China were collected."

Major comment 2:

Line 147 and figure 2, - clarification of the sampling method into these three groups is inappropriate. e.g. "medium-volume" could be used in "dilution sampling" -they considered different factors instead of different approaches of the same factor. The authors should rethink of the classification and reanalyze the part. A similar problem of classification is in the biomass burning part- Biomass boiler, FCE (field combustion experiment), LCS (laboratory chamber study), the first refers to the burning facility, while the other two are experiment methods. How could these three be in parallel here?

Response:

We thank the reviewer for bring this point to our attention. The classification of the sampling method is inappropriate. In fact, the source sampling method is varied with source type. For example, the sampling methods for vehicle emissions include direct measurement at exhaust pipe or by a dilution tunnel, tunnel experiment and sampling at underground parking lots. These sampling methods for vehicle emissions are different with dust, coal combustion and other sources.

We have re-classified the sampling methods for each primary source and updated Figure 2 in the manuscript.

Major comment 3:

The study analyzed sources in "coal combustion, industrial emissions, vehicle emissions, dust, cooking emissions, biomass burning" -some are based on fuel type, some are sectoral difference- this mixed classification should be corrected. e.g. "coal combustion and industrial emissions"- is coal burned in many industrial factories? What's the burning fuel in industrial combustion process?

Response:

We thank the reviewer for bring this important comment. Before addressing this issue, it should be noted that the source classification in source chemical profiles is different with that in emission inventories. The classification in emission inventories is based on sectoral difference, while the classification in source profiles is basically lies on their chemical nature. Thus, the source type is not always consistent with sectoral type particularly when the source profiles of two sectoral types are chemically similar.

The original "coal combustion and industrial emissions" is a generalized term that includes several sub-type sources. To make the source classification clearer to our readers, we now divided the "coal combustion and industrial emissions" into "Coal combustion emissions" and "Industrial process emissions". The "coal combustion emission" includes coal-fired power plants, coal-fired boilers and residential coal combustion. The "Industrial process emissions" denotes emissions from industrial production processes such as emissions from waste incineration, ceramic production, brick oven etc.

We have added a sentence in the Introduction section as follows:

"The main ubiquitous sources of atmospheric PM in China during the past three decades can be roughly divided into coal combustion sources (CC, with sub-type sources of coal-fired power plants, coal-fired boiler and residential coal combustion), vehicle exhaust (VE, gasoline and diesel engines), industrial processes emissions (IE), biomass burning (BB), cooking emissions (CE), fugitive dust (FD, with sub-type sources of soil fugitive dust, construction dust and road dust) and other localized specific sources."

Details on the source classification is available in the updated Figure 1.

Major comment 3:

Residential burning is a significant part of coal combustion, particularly in China. But the study did not have analysis and discussion on this. This important source should not be missed.

Response:

We thank the reviewer for highlighting this fact. In our previous manuscript, we

discussed the coal combustion and industrial emissions but without any discussion on residential coal combustion, which is an important source of ambient particulate matter, as suggested by the reviewer.

We have added the following paragraphs in Section 2.2.1 in response to this comment:

"In 2015, the total amount of coal consumption in mainland China is about 3970.14 Mt with a total of 93.47 Mt coal consumed in residential section. RCC is an important source of atmospheric PM in rural area, particularly in heating-season. Contrary to industrial furnaces and boilers, coal burned in household stoves has a significant impact on indoor and outdoor air quality in terms of its low thermal efficiency, incomplete combustion and the lack of air pollutant control devices. There are great efforts have been made to control air pollutants emitted from coal-fired power plants in China during past decades. It was reported that the emission factors of air pollutants for coal burned in household stoves are two more than two orders of magnitude higher than those burned in industrial boilers and power plants (Li et al., 2017), thus pollutants emitted from RCC have drawn great concern in recent years.

In general, coals can be classified as anthracite and bituminous coals in the forms of raw chunks and briquettes, burned with a movable brick or cast-iron stoves that has been used over centuries in China. There are many real-world measurements on particles emissions from RCC to investigate the emission nature. Most studies have rather placed focus on the emission factors than chemical composition as the emission factor of RCC has high uncertainty for a given air pollutant. The chemical characteristics of RCC profiles are varied greatly with sampling techniques. Three decades ago, Dai et al (1987) reported the averaged elemental profile of 15 RCC particle samples in Tianjin in 1985, with the use of Barco analyzer to cut fly ash (collected from the stack of RCC stove) into particles with aerodynamic diameter less than 12 μm. this poor sampling technique resulted in a high fraction of crustal elements in the chemical profile. The resuspension chamber has also been used to cut particle size from coal fly ash. However, the coal fly ash is not the particles emission from stack. Thus, the accuracy of RCC source profile has been improved until the dilution tunnel sampling method has been introduced into China. As shown in Fig.6, the fractions of crustal

elements (Mg, Al, Si, Ca, Ti) in the profile measured from coal ash are an order of magnitude higher than that in the RCC profile sampled by using dilution tunnel sampling method, while the fraction of sulfate, nitrate and OC are two to three orders of magnitude lower in coal ash $PM_{2.5}$.

Many efforts have been implemented in a national level to reduce pollutants emissions from RCC by introducing improved stoves and cleaner fuels since the 1990s, such as the China National Improved Stove Program. The highly efficient stove is reported likely has a reduced emission load. Given the limited available data, it is unable to compare the chemical profiles between the lowly and highly efficient stove at present. It is also reported that the emission factors of air pollutants from RCC varied widely because of the variations in coal type and property, stove type and burning condition. As shown in Fig. 7, $PM_{2.5}$ emission from the burning of chunk coals have a high fraction of OC, EC, sulfate, nitrate and ammonium, a low fraction of Na, Ca and K (K+) than the burning of honeycomb briquette coals. Generally, OC and sulfur is the predominate species in $PM_{2.5}$ emitted by RCC. It should be noted that, sulfate that is normally regarded as secondary species formed via oxidation processes in ambient air, accounted for ~8 to 38% of $PM_{2.5}$ mass emissions from RCC."

Minor comments:

**Comment 1 (original Line 166):** was TOT method used in EC/OC analysis?

Response:

Thanks for bring this point to our attention. Yes, the TOT method has also been used in OC/EC analysis.

We have edited the original sentence as follows:

"The total carbon (TC) mass is typically determined using thermal and thermal-optical methods. With the use of thermal/optical carbon analyzer, there are two widely used approaches to divide organic carbon (OC) and elemental carbon (EC) from TC, named DRI IMPROVE_A and NIOSH 5040, which are operationally defined by the time-temperature protocols, the OC/EC split approaches by optical

reflectance/transmittance."

**Comment 2 (original Line 180):** retene is also one widely used tracers for biomass burning, especially wood combustion.

Response:

Thanks. We have now added retene to the statement of the tracer of biomass burning.

**Comment 3 (original Line 263):** as seen in Figure 1 there is a significant number of coal emission studies, and even higher than the number of vehicle emissions. "rarely" might be inappropriate here.

Response:

We're noticed that this statement is inappropriate. We have deleted this sentence.

**Comment 4 (original Line 425):** may be good to provide some local studies on biomass burning emissions from indoor stoves and open burning.

Response:

In the revised version, the local studies on biomass burning emissions from indoor stoves and open burning have been added in this part as well as the response to comment 6.

**Comment 5 (original Line 427 and Figure 10):** as mentioned above, "biomass boiler" was not a "sampling method" because of the problem in such classification, the discussion from line 430 needs to be revised.

Response:

The classification of sampling method has been summarized and presented in Figure 2. We have updated Figure 10 (now Figure 12) to compare the profiles with different sampling and burning methods. Discussion about the comparison is available in Section 2.2.4 (now Section 2.2.5).

**Comment 6 (original Line 428):** is "biomass boiler" the boilers used in industry or

household? What about the home stoves?

Response:

The biomass boiler here is the bio-fuel boiler for industry purpose. In the revised MS, more descriptions including home stoves emissions were added as follows:

"Bio-fuels are usually burned in three ways in China, that is open burning (OB), residential stove combustion (RSC), and biofuel boiler burning (BBB). At present, there are two popular ways in the measurements of biomass burning: field combustion experiment (FCE) and laboratory combustion simulation (LCS) (Hays et al., 2005; Li et al., 2014a; Sanchis et al., 2014; De Zarate et al., 2000). Fig. 12 summarizes the biomass burning source profiles of $PM_{2.5}$ from three burning styles obtained in China. The samples of biomass boiler exhaust are obtained by resuspension sampling method. The main components in the profiles of biomass burning are OC, EC, $K^+$ (K), $Cl^-$ and Ca (Fig. 12). The fraction of EC is 4.2 times higher in BBB than RSC, which is potentially due to the uneven mixing of the air in the biomass boiler that easy to make straw burning in anaerobic condition (Tian et al., 2017). The high EC emissions can also happen if high temperature flaming burning condition dominant in the BBB. The oxygen content is relatively sufficient in OB, which leads to relatively higher OC emission. The fraction of Ca was higher in BBB exhaust than OB (Fig. 12)."

**Comment 7 (original Line 433-434):** high EC or BC emissions are usually due to the domination of high temperature flaming burning conditions.

Response:

We agree with the reviewer that the high EC emission are likely caused by the high temperature flaming burning conditions, however we don't have much more evidence to prove that is true in bio-fuel boilers. The original sentence has been modified as follows:

"The fraction of EC is significant higher in the bio-fuel boiler exhaust than the laboratory combustion simulation, which is potentially due to the uneven mixing of the air in the biomass boiler that easy to make straw burning in anaerobic condition (Tian et al., 2017). The high EC emissions can also happen if high temperature flaming

burning condition dominant in the bio-fuel boiler."

**Comment 8 (original Line 437-438):** it seems that the authors only considered open burning when referring to"FCE".

Response:

The discussion about the sampling method for biomass burning is based on the Section 2.1 (Figure 2), including dilution tunnel sampling for combustion chamber simulation, stove combustion, open field burning and bio-fuel boiler, and direct plume sampling in open field.

**Comment 9 (original Lines 441-443):** a recent study showed that for open biomass burning, emission factors of most air pollutants from field measurements and simulated chamber studies in laboratory are comparable.

Response:

Interesting, we agree that this would be a possibility, but the data presented here is not support that conclusion.

**Comment 10 (original Line 446):** also due to high combustion temperature and flaming dominance burning conditions.

Response:

It is possible for the formation of EC. We now have edited the original sentence as follows:

"For specific components emissions from the biomass burning, EC emissions from firewood combustion was found to be the highest, which is likely due to the high combustion temperature and flaming dominance burning condition, and the higher content of lignin in wood (Tang et al., 2014), since lignin facilitates the formation of black carbon (Wiinikka and Gebart, 2005)."

**Comment 11 (original Line 527):** the reference "Yan et al., 2017" is missing.

Response:

Thanks, it has been added to the reference list.

**Comment 12 (original Line 555-556):** as mentioned above the biomass boilers are burning facilities while the other two are sampling/experiment methods.

Response:

We have modified the classification of source sampling for biomass burning source. This sentence has been edited now.

**Comment 13 (original Line 600**): please consider enlarging and improving the resolution Giving different burning conditions and different fuel properties even for the same type of fuel, within the current compiled database, it is interesting to look into the spatial difference in source profiles for the same source type.

Response:

The original Figure 14 has been updated to a high resolution and clearer version (now Figure 16 in the revised MS). For the spatial difference, we have added a paragraph with more descriptions on the profiles detected in different areas to Section 2 in the revised version as follows:

"These published profiles were detected in different parts of China. In eastern China, there are published profiles of 35 CC, 14 IE, 14 VE, 18 BB, and 2 CE; in northern China, there are published profiles of 16 CC, 23 IE, 9 VE, 8 BB and 13 CE; in western China, there are only profiles of 20 CC; in southern China, there are published profiles of 10 VE and 10 CE; in central China, there are published profiles of 17 BB. For example, the profiles of residential coal combustion are mainly detected in the regions that have obvious activities of residential coal burning, such as the northern and western China. The region of different parts of China was defined by Zhu (2018)."

**Comment 14:** Generally, the manuscript is understandable but could be improved after

a language edit and polish. Please go through the manuscript and check for grammars and spelling errors for example, Line 79, "source measurement is actually it is time to . . ." -the sentence is incomplete. Please check and revise.

Response:

The English of the revised MS has been improved by a native speaker. We have checked the revised MS several times to correct the grammar errors.

---

## Author Comment (AC2) · 5 Dec 2018

**Reviewer #4:**

This manuscript presented detailed characteristics of the main primary source profiles of PM in China. The conclusions here can provide clear evidences for the source apportionment and environmental management. Reviews of evolutions of sampling methods, chemical analytical methods and source profiles were also given. Besides, the authors also proposed future requirements for the development of source profiles in China. However, some descriptions in the manuscript need to be further improved, and more tracer characteristics of the sources need to be discussed. This manuscript can be considered for publication in Atmospheric Chemistry and Physics after major revision.

Specific Comments:

**Comment 1 (original Lines 79-80):** rewrite the sentence.

Response:

This whole paragraph has been revised now.

**Comment 2 (original Lines 96-97):** except the key words listed in the manuscript, have you ever considered other keywords, such as coal combustion, industrial emissions... for the literature research?

Response:

Thanks for your suggestion. To avoid missing any papers, we now have collected the published papers using the following strategy with more source-related key words.

"The search keywords depend on source category. The following keywords for each source were used individually or in combination. As for CC sources, the key words are "coal combustion/coal burning/coal-fired boiler/coal-fired power plant/residential coal" and "source profile/chemical profile/particle composition". The key words for other sources are shown as follows. IE: "industrial emission" and "source profile/chemical profile/particle composition"; VE: "vehicle emission/exhaust emission/traffic emission/diesel engine/truck emission/gasoline engine/on-road vehicle/tunnel experiment/chassis dynamometer/portable emission measurement system" and "source profile/chemical profile/particle composition"; CE: "cooking emission" and "source

profile/chemical profile/particle composition"; BB: "biomass burning/bio-fuel boiler" and "source profile/chemical profile/particle composition"; FD: "soil/fugitive dust/crustal material/construction dust/road dust" and "source profile/chemical profile/particle composition".".

We have added the details of literature search strategy to the Introduction section.

**Comment 3:** Section 2.2.1, it was shown in Figure 1 that residential coal combustion contributed 20 literatures, but it has been completely neglected in this section. So far as I know, PM emitted from residential coal combustion is quite different from IBW and PPW. Please give further discussion about residential coal combustion.

Response:

This is a great point that was also brought up by Reviewer #2. Indeed, the residential coal combustion (RCC) source is an important source of atmospheric particulate matter. We have added the following paragraphs in Section 2.2.1 in response to this comment:

"In 2015, the total amount of coal consumption in mainland China is about 3970.14 Mt with a total of 93.47 Mt coal consumed in residential section. RCC is an important source of atmospheric PM in rural area, particularly in heating-season. Contrary to industrial furnaces and boilers, coal burned in household stoves has a significant impact on indoor and outdoor air quality in terms of its low thermal efficiency, incomplete combustion and the lack of air pollutant control devices. There are great efforts have been made to control air pollutants emitted from coal-fired power plants in China during past decades. It was reported that the emission factors of air pollutants for coal burned in household stoves are two more than two orders of magnitude higher than those burned in industrial boilers and power plants (Li et al., 2017), thus pollutants emitted from RCC have drawn great concern in recent years.

In general, coals can be classified as anthracite and bituminous coals in the forms of raw chunks and briquettes, burned with a movable brick or cast-iron stoves that has been used over centuries in China. There are many real-world measurements on particles emissions from RCC to investigate the emission nature. Most studies have

rather placed focus on the emission factors than chemical composition as the emission factor of RCC has high uncertainty for a given air pollutant. The chemical characteristics of RCC profiles are varied greatly with sampling techniques. Three decades ago, Dai et al (1987) reported the averaged elemental profile of 15 RCC particle samples in Tianjin in 1985, with the use of Barco analyzer to cut fly ash (collected from the stack of RCC stove) into particles with aerodynamic diameter less than 12 μm. this poor sampling technique resulted in a high fraction of crustal elements in the chemical profile. The resuspension chamber has also been used to cut particle size from coal fly ash. However, the coal fly ash is not the particles emission from stack. Thus, the accuracy of RCC source profile has been improved until the dilution tunnel sampling method has been introduced into China. As shown in Fig. 6, the fractions of crustal elements (Mg, Al, Si, Ca, Ti) in the profile measured from coal ash are an order of magnitude higher than that in the RCC profile sampled by using dilution tunnel sampling method, while the fraction of sulfate, nitrate and OC are two to three orders of magnitude lower in coal ash $PM_{2.5}$.

Many efforts have been implemented in a national level to reduce pollutants emissions from RCC by introducing improved stoves and cleaner fuels since the 1990s, such as the China National Improved Stove Program. The highly efficient stove is reported likely has a reduced emission load. Given the limited available data, it is unable to compare the chemical profiles between the lowly and highly efficient stove at present. It is also reported that the emission factors of air pollutants from RCC varied widely because of the variations in coal type and property, stove type and burning condition. As shown in Fig. 7, $PM_{2.5}$ emission from the burning of chunk coals have a high fraction of OC, EC, sulfate, nitrate and ammonium, a low fraction of Na, Ca and K ($K^+$) than the burning of honeycomb briquette coals. Generally, OC and sulfur is the predominate species in $PM_{2.5}$ emitted by RCC. It should be noted that, sulfate that is normally regarded as secondary species formed via oxidation processes in ambient air, accounted for ~8 to 38% of $PM_{2.5}$ mass emissions from RCC."

**Comment 4 (original Figure 4):** Only OC, EC and Ca were described in line 221-225,

however, other components such as NO3-, Cl-, and Ca2+ also varied significantly between EP and EBCC, please give more descriptions; Only SO42-, Ca and OC were described in line 226-234, how about other components such as NH4+, Na+ and Cl-? Ca and Ca2+ showed opposite tendencies between DD and WFGD, please give reasonable explanation.

Response:

More descriptions have been added in the revised MS in response to this comment. The reason for Ca and $Ca^{2+}$ showed opposite tendencies between DD and WFGD probably is the different solubility of Ca compounds between them. For WFGD, Ca mainly exists as $CaSO_4$ that has low solubility, while for DD, Ca probably exists as the compounds with higher solubility. This is an inference that needs more investigation in the future.

**Comment 5 (original Lines 246-247):** it is said that Si, OC and EC from RSM are significantly higher than DTSM, however, it is showed from Figure 5 that Si and OC had almost the same medium value and average value for RSM and DTSM, I do not agree about this conclusion.

Response:

Thanks for the comment. We've double-checked the data used in Figure 5 (now Figure 4 in the revised MS), and found some mistakes in our original data treatment. The updated Figure are shown in the revised MS as Figure 4. The description of this Figure is also updated as follows:

"For RSM, the crustal elements (Si) and EC are significantly higher than DTSM. The $SO_4^{2-}$ fraction of DTSM is significantly higher than RSM, reaching 0.1643 g/g. And V, Cr, Mn, Co, Ni, Cu, Zn, Pb and other trace metal fractions are strongly enriched in DTSM, which is 1.7 to 60.7 times that in RSM".

**Comment 6 (original Lines 263):** a total of 71 literatures are showed in Figure 1, why said "rarely" here?

Response:

Indeed, this statement is inappropriate. We have deleted this sentence.

**Comment 7:** Mn and Pb in Figure 7 showed characteristics only can correspondence with gasoline vehicles in 2015 (Figure 8). So, are the data in Figure 7 and Figure 8 from the same data source? And which year? Can they represent the vehicle emissions? Are the vehicles in Figure 8 gasoline vehicles?

Response:

It is not the same data source for Figure 7 and 8 (now Figure 9 and 10). In Figure 7 (now Figure 9), the profiles from the same sampling method for different vehicle types were compared to compare the difference among these types. While in Figure 8 (now Figure 10), Mn, Pb, and $SO_4^{2-}$ in the profiles from different years in the past three decades were reviewed. Due to the limited information of the original citations, we only confirmed that the some profiles used in Figure 8 (now Figure 10) are a mixture of different vehicle types and the vehicles were not only gasoline vehicles. We have searched all the possible literatures for this topic, and we believe the variation trend of these species could be represented by these profiles from the citations.

**Comment 8 (original Line 337):** a space is missed between "also" and "varied".

Response:

Thanks for your reminder.

**Comment 9 (original Line 396):** full names of SD and RD should be given for the first appearance.

Response:

Thanks. The full names of SD and RD are given for the first appearance as soil dust and road dust, respectively.

**Comment 10 (original Lines 441-443):** rewrite the sentence.

Response:

We think the previous statement is incorrect so we delete it in the updated manuscirpt.

**Comment 11 (original Line 434):** it was mentioned that the relatively sufficient oxygen content could help for the OC formation, and in Line 449, the complete combustion was considered can reduce the production of OC, please give reasonable explanation.

Response:

Thanks for your comment. The statement in the Line 449 of the original MS is an inference that was lack of experimental or theory basis from the original reference. In the revised version, this sentence has been deleted.

**Comment 12 (original Line 452):** check the spellings.

Response:

Thanks for your reminder. We have corrected the caption of previous Figure 10.

**Comment 13:** Section 2.2, characteristics of PM from several sources were discussed here, however, in my opinion, it is better to give more tracers or distinguished features of each source, which can make it easier to identify different sources.

Response:

Thank you for your insightful comment. The profiles discussed in this paper were mainly consisted of routine chemical components. From our experience, a single routine species is not always enough to be used as a tracer that fully represents for a certain source when performing the CMB or PMF calculations for source apportionment. For example, OC could be the tracer of coal combustion, vehicle exhaust, or biomass burning. In most cases, the tracer of sources depends on the species used for fitting. It is a combination of chemical species rather than a single species. Thus, we do not add a table of tracers for these routine species in the revised MS.

---

## Author Comment (AC3) · 5 Dec 2018

Reviewer #3:

The authors investigated the evolution of primary source profiles of PM in China between 1987-2017. They reviewed a total of 3244 chemical profiles, assessed their uncertainties, and conducted a cluster analysis to analyze the heterogeneity across different source categories. There are many studies in literature that have summarized the characteristics of PM source profiles in China. Compared to the previous studies, the method used here is not novel, and I don't see much scientific significance in this paper though it summarized plenty of data and did some analysis. The paper is not well written and needs lots of editing. My major comments are as follows.

**Comment 1:** After reading the title, I expected the evolution of source profile with time was one research focus. However, the paper only analyzed the evolution of source profiles from vehicle emissions. I suggest the authors provide more discussions on other important sources (e.g., coal combustion and industrial emissions) if possible.

Response:

We thank the reviewer for highlighting this fact. The main point of this review work is to characterize the evolution of the main primary source profiles in China during the last three decades. To fully address this issue, we have added a deep-discussion of the source profiles to the revised MS.

As for coal combustion emissions, the source profiles have changed greatly with the advancement of the source sampling method since the 1980s. Previously, researchers have used the Barco particle size analyzer to obtain particles with aerodynamic diameter less than 12 μm as particle samples ($PM_{12}$) by cutting coal fly ash, which was collected from the stacks of industrial coal boilers and domestic stoves as the emission particle samples of coal combustion sources (Dai et al., 1987). With the development and application of resuspension sampling technique in China after the 1990s (Chow et al., 1994; Ho et al., 2003), the collected coal ash can suspend in the resuspension chamber and then sampled by ambient particle sampler. However, both of these two methods using the coal fly ash to represent the emissions from stationary coal combustion sources, which is not the real emissions in nature. Until the dilution tunnel sampling technique appears after the year of 2000, the particle can be sampled by using

isokinetic sampling method in the chimney flue. The composition of coal combustion sources varied with the sampling methods as expected. The fraction of crustal elements in coal ash deduced profiles is higher than that in profiles associated with dilution tunnel sampling, while coal ash deduced profiles have low fractions of sulfate. This effect resulted from sampling method works for all subtype sources of coal combustion, as shown in Figure 4 and 6.

We have added the following statements at the following locations within the manuscript to reflect our response to this issue:

1. Addition to Section 2.1:

"Since the 1970s, dilution tunnel sampling method (DTSM) has been developed to originally obtain source samples from vehicle emissions that could be close to the real compositions from the sources. Subsequently, various dilution tunnels have been developed with different tunnel materials, resident time, dilution ratios, diameter of effective mixing lengths to collect particles emissions from stationary sources. The development and application of such technique in China was after 2000, while it has been widely used nowadays."

2. Addition to Section 2.2.1:

"The chemical characteristics of RCC profiles are varied greatly with sampling techniques. Three decades ago, Dai et al (1987) reported the averaged elemental profile of 15 RCC particle samples in Tianjin in 1985, with the use of Barco analyzer to cut fly ash (collected from the stack of RCC stove) into particles with aerodynamic diameter less than 12 $\mu$m. this poor sampling technique resulted in a high fraction of crustal elements in the chemical profile. The resuspension chamber has also been used to cut particle size from coal fly ash. However, the coal fly ash is not the particles emission from stack. Thus, the accuracy of RCC source profile has been improved until the dilution tunnel sampling method has been introduced into China. As shown in Fig. 7, the fractions of crustal elements (Mg, Al, Si, Ca, Ti) in the profile measured from coal ash are an order of magnitude higher than that in the RCC profile sampled by using dilution tunnel sampling method, while the fraction of sulfate, nitrate and OC are two to three orders of magnitude lower in coal ash $PM_{2.5}$."

**Comment 2:** Abstract. The authors mentioned "the most complicated profiles are likely attributed to coal combustion and industrial emissions." (Line 17). This is well recognized thus not appropriate to repeat it in the abstract. Please focus on the main findings of this study. For example, the results of cluster analysis should be summarized in the abstract.

Response:

We thank the reviewer for bringing this suggestion. We have edited the Abstract in the updated manuscript.

**Comment 3:** Introduction. The introduction part presents weak literature reviews. A literature review is much more than a descriptive list of materials available.

Response:

Thanks for your suggestion. Our revision to this section is included in the following bulleted list:

1. Added sentences to the end of the first sentence of the second paragraph in the Introduction section to critical review the development of source profiles in China:

"The time evolution of source profiles is partly determined by the source apportionment techniques. In general, the receptor model was developed based on the assumption of mass conservation. A mass balance equation represents that the measured particle mass can be regarded as the linear sum of the mass of all chemical components contributed from several sources. Initially, the mass balance equations were deployed for a couple of specific elements and source types in America. Elements, ions and carbon materials gradually tend to be the routine chemical species in the source apportionment of PM. With the development of advanced sampling and chemical analysis techniques, more valuable information, such as organic compounds, isotopic measurement of radiocarbon, sulfur and nitrogen and high-resolution aerosol mass spectra and particle size distribution etc., have been explored to further expand the existing or new profiles. This information has been proved to provide source specificity

capable of being incorporated into receptor models as new markers, constraining source contributions, and developing new models. For example, Dai et al (2019) developed a size-resolved CMB approach for source apportionment of PM based on the size profiles of sources. The new valuable information gives significant possibilities to source apportionment models to obtain more precise and reliable results.

2. Added a paragraph to review the current state of source profiles in China:

"Since the 1980s, source profile studies were initially implemented in China (Dai et al., 1987). During the past three decades, hundreds of source profiles have been achieved across China. These profiles covered more than forty cities and several source types. The main ubiquitous sources of atmospheric PM in China during the past three decades can be roughly divided into coal combustion sources (CC, with sub-type sources of coal-fired power plants, coal-fired boiler from industry and residential coal combustion), vehicle exhaust (VE, gasoline and diesel engines), industrial processes emissions (IE), biomass burning (BB), cooking emissions (CE), fugitive dust (FD, with sub-type sources of soil fugitive dust, construction dust and road dust) and other localized specific sources. These available profiles have filled the gap of the knowledge of source compositions and provided effective markers for the source apportionment studies. However, the current state and issues of pre-existing primary source profiles of PM in China are still unclear, it is time to overview these source profiles along the time line and add more profile knowledge to the atmospheric research community."

**Comment 4:** Method. It is not clear to me how the authors selected the source profile that is of acceptable quality. What is the criteria for inclusion or exclusion of a profile from a literature? It is important that the method part is self-contained and clear enough for audiences to reproduce the given results.

Response:

This is a very valid point that was also brought by reviewers #2 and #4. To address this issue, we have clarified the literature search strategy as follows.

"To collect the potential published data related to source profiles, a two-round literature

search work covering literature from 1980 to 2018 was done in this work. In the first round of searching, two authors are responsible for the same source to ensure every source category has been searched twice independently. The search keywords depend on source category. The following keywords for each source were used individually or in combination. As for CC sources, the key words are "coal combustion/coal burning/coal-fired boiler/coal-fired power plant/residential coal" and "source profile/chemical profile/particle composition". The key words for other sources are shown as follows. IE: "industrial emission" and "source profile/chemical profile/particle composition"; VE: "vehicle emission/exhaust emission/traffic emission/diesel engine/truck emission/gasoline engine/on-road vehicle/tunnel experiment/chassis dynamometer/portable emission measurement system" and "source profile/chemical profile/particle composition"; CE: "cooking emission" and "source profile/chemical profile/particle composition"; BB: "biomass burning/bio-fuel boiler" and "source profile/chemical profile/particle composition"; FD: "soil/fugitive dust/crustal material/construction dust/road dust" and "source profile/chemical profile/particle composition". Papers and dissertations in Chinese on China National Knowledge Infrastructure (CNKI) and papers in English on the web of science were searched using above keywords, respectively. The duplicated paper was then double-checked and excluded. The papers with topic related to source profiles but without providing any information of real-measured sources were also excluded. For example, papers reported source apportionment results with the use of PMF and CMB but without reporting local profiles were not taken into account. As a result, a total of 193 papers have been collected from these efforts. In the second round of searching, the valid papers with available source profile data and detailed source sampling and chemical analysis methods were counted and used for post-analysis. Finally, a total of 456 published source profiles since the 1980s across China were collected. "

We have also added the details of literature search to the Introduction section in response to this comment.

**Comment 5:** Section 2.3. The title need to be reconsidered since this section contains

the analysis using the coefficient of variation as well.

Response:

This is an astute observation by the reviewer. We have changed the title of section 2.3 as "Statistic analysis of the source categories". We want to mention that the statistical methods used here are aim to objectively test the homogeneity of sources for the given (subjectively known) source categories.

**Comment 6:** As a significant source, residential coal combustion is missed in the paper. Please provide more discussions.

Response:

This is a great point that was also brought up by Reviewers #2 and #4. We agree with the reviewer that the residential coal combustion (RCC) source is an important source of atmospheric particulate matter. We have added the following paragraphs in Section 2.2.1 in response to this comment:

"In 2015, the total amount of coal consumption in mainland China is about 3970.14 Mt with a total of 93.47 Mt coal consumed in residential section. RCC is an important source of atmospheric PM in rural area, particularly in heating-season. Contrary to industrial furnaces and boilers, coal burned in household stoves has a significant impact on indoor and outdoor air quality in terms of its low thermal efficiency, incomplete combustion and the lack of air pollutant control devices. There are great efforts have been made to control air pollutants emitted from coal-fired power plants in China during past decades. It was reported that the emission factors of air pollutants for coal burned in household stoves are two more than two orders of magnitude higher than those burned in industrial boilers and power plants (Li et al., 2017), thus pollutants emitted from RCC have drawn great concern in recent years.

In general, coals can be classified as anthracite and bituminous coals in the forms of raw chunks and briquettes, burned with a movable brick or cast-iron stoves that has been used over centuries in China. There are many real-world measurements on particles emissions from RCC to investigate the emission nature. Most studies have rather placed focus on the emission factors than chemical composition as the emission

factor of RCC has high uncertainty for a given air pollutant. The chemical characteristics of RCC profiles are varied greatly with sampling techniques. Three decades ago, Dai et al (1987) reported the averaged elemental profile of 15 RCC particle samples in Tianjin in 1985, with the use of Barco analyzer to cut fly ash (collected from the stack of RCC stove) into particles with aerodynamic diameter less than 12 μm. this poor sampling technique resulted in a high fraction of crustal elements in the chemical profile. The resuspension chamber has also been used to cut particle size from coal fly ash. However, the coal fly ash is not the particles emission from stack. Thus, the accuracy of RCC source profile has been improved until the dilution tunnel sampling method has been introduced into China. As shown in Fig.6, the fractions of crustal elements (Mg, Al, Si, Ca, Ti) in the profile measured from coal ash are an order of magnitude higher than that in the RCC profile sampled by using dilution tunnel sampling method, while the fraction of sulfate, nitrate and OC are two to three orders of magnitude lower in coal ash $PM_{2.5}$.

Many efforts have been implemented in a national level to reduce pollutants emissions from RCC by introducing improved stoves and cleaner fuels since the 1990s, such as the China National Improved Stove Program. The highly efficient stove is reported likely has a reduced emission load. Given the limited available data, it is unable to compare the chemical profiles between the lowly and highly efficient stove at present. It is also reported that the emission factors of air pollutants from RCC varied widely because of the variations in coal type and property, stove type and burning condition. As shown in Fig. 7, $PM_{2.5}$ emission from the burning of chunk coals have a high fraction of OC, EC, sulfate, nitrate and ammonium, a low fraction of Na, Ca and K (K+) than the burning of honeycomb briquette coals. Generally, OC and sulfur is the predominate species in $PM_{2.5}$ emitted by RCC. It should be noted that, sulfate that is normally regarded as secondary species formed via oxidation processes in ambient air, accounted for ~8 to 38% of $PM_{2.5}$ mass emissions from RCC."

**Comment 7 (original Line 184):** the description of VOCs source profiles seems not quite related to the topic of this paper.

Response:

Thanks for bring this comment to our attention. We have deleted the text associated with VOCs in the updated manuscript.

**Comment 8 (original Line 246 and Figure 5):** please check the figure and raw data if Si and carbon components for RSM are significantly higher than DTSM.

Response:

Thanks for your comment. We've double-checked the data used in previous Figure 5 (now Figure 4 in the revised MS), and found some mistakes in the original data treatment. The updated Figure are shown in the revised MS as Figure 4. The description of this Figure is also updated as follows:

"For RSM, the crustal elements (Si) and EC are significantly higher than DTSM. The $SO_4^{2-}$ fraction of DTSM is significantly higher than RSM, reaching 0.1643 g/g. And V, Cr, Mn, Co, Ni, Cu, Zn, Pb and other trace metal fractions are strongly enriched in DTSM, which is 1.7 to 60.7 times that in RSM".

**Comment 9:** Figure 11, please clarify the information of the chemical profile given here, i.e., is it an average profile or related to a specific cooking style?

Response:

This is an average profile from all reported cooking styles. We have edited the caption of the original Figure 11 (now Figure 13).

**Comment 10:** Many syntax and spelling errors in the text. For example, Line 33, "While the profiles of road dust and soil dust……"; Line 307, "Given that there are many factors……".

Response:

The English of the revised MS has been improved by a native speaker. We have checked the revised MS several times to correct the grammar errors.

---

## Author Comment (AC4) · 5 Dec 2018

Reviewer #4(5):

The knowledge of source profiles in China is significantly inadequate. In this manuscript, the authors aimed to review the characteristics and evolution of source profiles in China from 1987 to 2017, which would provide very necessary information for source apportionment and evaluation of health effect from different sources. But, ACP as one of the high level paper at area of atmosphere research, the manuscript should be revised largely to deep discuss the evolution of source profiles. The latest version was considered without compact structure and profound discussion. I would like to review again after some major revision done.

**Major comments 1:**

Although it was reported that 3244 chemical profiles was discussed in this study, the authors should consider how could those database of profiles be used by other researchers? The latest version couldn't show the huge amount of data. It seems that some table for profiles were better than figure.

Response:

We are trying to make the database available to our research community through an easy access App, which is still under development. At present, we have made tables in the supplemental material to present profiles data.

**Major comments 2:**

The structure of manuscript was not compact, etc. part 2.1. The manuscript should be written more logic.

Response:

We have almost totally rewritten the Section 1 and Section 2.1 to make it more logic and clear enough. Our manuscript provides the details.

**Major comments 3:**

One of the most important aims of this manuscript is to evolution the changes of profiles from 1987 to 2017. However, some profiles like coal combustion couldn't show this trend. It should be better discussion from some aspects like source profiles variation

from different years, processing and sampling methods.

Response:

This is a valid point that was also brought up by Reviewer #3. The main point of this review work is to characterize the evolution of the main primary source profiles in China during the last three decades. To fully address this issue, we have added a deep-discussion of the source profiles to the MS.

As for coal combustion emissions, the source profiles have changed greatly with the advancement of the source sampling method since the 1980s. Previously, researchers have used the Barco particle size analyzer to obtain particles with aerodynamic diameter less than 12 µm as particle samples ($PM_{12}$) by cutting coal fly ash, which was collected from the stacks of industrial coal boilers and domestic stoves as the emission particle samples of coal combustion sources (Dai et al., 1987). With the development and application of resuspension sampling technique in China after the 1990s (Chow et al., 1994; Ho et al., 2003), the collected coal ash can suspend in the resuspension chamber and then sampled by ambient particle sampler. However, both of these two methods using the coal fly ash to represent the emissions from stationary coal combustion sources, which is not the real emissions in nature. Until the dilution tunnel sampling technique appears after the year of 2000, the particle can be sampled by using isokinetic sampling method in the chimney flue. The composition of coal combustion sources varied with the sampling methods as expected. The fraction of crustal elements in coal ash deduced profiles is higher than that in profiles associated with dilution tunnel sampling, while coal ash deduced profiles have low fractions of sulfate. This effect resulted from sampling method works for all subtype sources of coal combustion, as shown in Figure 4 and 6.

We have added the following statements at the following locations within the manuscript to reflect our response to this issue:

1.  Addition to Section 2.1:

"Since the 1970s, dilution tunnel sampling method (DTSM) has been developed to originally obtain source samples from vehicle emissions that could be close to the real compositions from the sources. Subsequently, various dilution tunnels have been

developed with different tunnel materials, resident time, dilution ratios, diameter of effective mixing lengths to collect particles emissions from stationary sources. The development and application of such technique in China was after 2000, while it has been widely used nowadays."

2. Addition to Section 2.2.1:

"The chemical characteristics of RCC profiles are varied greatly with sampling techniques. Three decades ago, Dai et al (1987) reported the averaged elemental profile of 15 RCC particle samples in Tianjin in 1985, with the use of Barco analyzer to cut fly ash (collected from the stack of RCC stove) into particles with aerodynamic diameter less than 12 μm. this poor sampling technique resulted in a high fraction of crustal elements in the chemical profile. The resuspension chamber has also been used to cut particle size from coal fly ash. However, the coal fly ash is not the particles emission from stack. Thus, the accuracy of RCC source profile has been improved until the dilution tunnel sampling method has been introduced into China. As shown in Fig. 7, the fractions of crustal elements (Mg, Al, Si, Ca, Ti) in the profile measured from coal ash are an order of magnitude higher than that in the RCC profile sampled by using dilution tunnel sampling method, while the fraction of sulfate, nitrate and OC are two to three orders of magnitude lower in coal ash $PM_{2.5}$."

**Major comments 4:**

It seems that some source profiles in China were not included in this review. I suggested that the authors should search more carefully. For example, the amount of source profiles for diesel emission published already would never be so small.

Response:

This is an important point that was also brought up by Reviewers #2 and #4. We now have searched the literatures again based on a two-round paper search work and using more source-related key words. Finally, a total of 456 published source profiles since the 1980s across China were collected. Details on the literature search is available in the Introduction section.

**Major comments 5:**

Many sentences were long and complicated, which were hard to understand. Some short and simple sentence should be better (etc. lines50-53; 59-63; 79-80;124-126…).

Response: In the revised version, some long sentences are shortened to make the express simple and clear. There are no unnecessary long sentences in the revised MS.

**Major comments 6:**

It would be better that give some review about source profiles with organic matter, isotopes and size distribution (according to line 71-75).

Response:

Thanks for your suggestion. We have added the review of the source profiles to the Introduction section in terms of the physicochemical nature (lines 74-86):

"Initially, the mass balance equations were deployed for a couple of specific elements and source types in America. Elements, ions and carbon materials are gradually tend to be the routine chemical species in the source apportionment of PM. With the development of advanced sampling and chemical analysis techniques, more valuable information, such as organic compounds, isotopic measurement of radiocarbon, sulfur and nitrogen and high-resolution aerosol mass spectra and particle size distribution etc., have been explored to further expand the existing or new profiles. This information has been proved to provide source specificity capable of being incorporated into receptor models as new markers, constraining source contributions, and developing new models. For example, Dai et al (2019) developed a size-resolved CMB approach for source apportionment of PM based on the size profiles of sources. The new valuable information gives significant possibilities to source apportionment models to obtain more precise and reliable results."

Minor comments:

**Comment 1 (Original Lines 65-66):** add more typical research about source profiles. Line 71-73: add references.

Response:

Thanks for your suggestion. More references have been added to support the statements.

**Comment 2 (Original Lines 96-100):** changed the sentence into passive "…were used as the key words…", and delete "searching for papers and dissertations".

Response:

This paragraph has been modified now.

**Comment 3 (Original Line 100):** delete "the source profile data were compiled".

Response:

It has been deleted now.

**Comment 4:** There is not shown the size distribution in Figure 1 (lines 106-108).

Response:

Figure 1 has been updated with counts in particle size.

**Comment 5:** How about the source profiles detected in different areas? (part 2) (give the data marked in map is better)

Response:

It is a wonderful suggestion, and we try to draw such map in the modified version, but we found it is difficult to demonstrate all the information in one map. To address this point, we have added more descriptions on the profiles detected in different areas to Section 2.1 in the revised version as follows:

"These published profiles were detected in different parts of China. In eastern China, there are published profiles of 35 CC (excluded residential coal combustion), 14 IE, 14 VE, 18 BB, 2 CE and 14FD; in northern China, there are published profiles of 16 CC, 23 IE, 9 VE, 8 BB 13 CE and 62FD; in western China, there are only profiles of 20 CC; in southern China, there are published profiles of 10VE, 10CE, and 5FD; in central China, there are published profiles of 17 BB."

**Comment 6 (Original Line 120):** is it source profile research not source apportionment research? What the meaning of catch?

Response: It is source apportionment research. *Catch* here means "match".

**Comment 7:** Variations of sampling methods during different periods were more important (Figure 2).

Response:

We agree with the reviewer that the sampling method played an important role in the variation of source profiles. Here we take coal combustion source as an example, the source profiles have changed greatly with the advancement of the source sampling method since the 1980s. Previously, researchers have used the Barco particle size analyzer to obtain particles with aerodynamic diameter less than 12 µm as particle samples ($PM_{12}$) by cutting coal fly ash, which was collected from the stacks of industrial coal boilers and domestic stoves as the emission particle samples of coal combustion sources (Dai et al., 1987). With the development and application of resuspension sampling technique in China after the 1990s (Chow et al., 1994; Ho et al., 2003), the collected coal ash can suspend in the resuspension chamber and then sampled by ambient particle sampler. However, both of these two methods using the coal fly ash to represent the emissions from stationary coal combustion sources, which is not the real emissions in nature. Until the dilution tunnel sampling technique appears after the year of 2000, the particle can be sampled by using isokinetic sampling method in the chimney flue. The composition of coal combustion sources varied with the sampling methods as expected. The fraction of crustal elements in coal ash deduced profiles is higher than that in profiles associated with dilution tunnel sampling, while coal ash deduced profiles have low fractions of sulfate. This effect resulted from sampling method works for all subtype sources of coal combustion, as shown in Figure 4 and 6 To fully address this issue, we have added a deep-discussion of the source profiles to the Section 2.2.1. Please see response to comment 1 from Reviewer #3 for details.

**Comment 8 (Line 181):** check the format of comma.

Response:

It has been revised.

**Comment 9**. Check the format of citation all of the manuscript.

Response:

We have carefully checked the format of the citations across the manuscript.

**Comment 10**. Line 333-336: I wondered that the precursors of NO32- and NH4+ were VOC?

Response: The original statement in the MS is incorrect and we modified the sentence as '$NH_4^+$ and $NO_3^-$ in chemical profiles obtained by DSM are lower than that of SDSM, probably because their precursors are still in the gaseous state when the samples were collected at a higher temperature by DSM'.

**Comment 11**. Check line 337, lines 339-340. Some sentence seems were copy by other places, which color was different with the normal.

Response: We have modified the sentences with different colors in the revised MS according to this comment.

**Comment 12**. Figure 8: please explain why the trend of Mn was increasing after 2005?

Response: The increase of Mn after 2005 may due to the sample differences, such as sampling locations, vehicle types and age. It should be noted that the content of Mn remained a low level among $10^{-4}$~$10^{-3}$ after 2005, while such content was between $10^{-2}$~$10^{-3}$ before 2005. The evolution trend of Mn in the profile of VE decreased to a rather low level after 2005.

**Comment 13**. It would be better that some tables or figure to comparing the difference of source profiles between China and EPA (lines 360-370).

Response:

We have compared some profiles between China and EPA Speciate database and

presented the details in the text in the updated MS.

**Comment 14**. Figure 14 was hard to read.

Response: According to this comment, the original Figure 14 has been updated to a high resolution and clearer version (Figure 16 in the revised MS).

**Comment 15**. It would be better to give the fractions of typical species to PM for each profile, which could be evaluated whether the dominant species could be used as biomarker.

Response: In the revised version, a table contained the detailed information of the published profiles was added to Supplemental materials. In this table, the fractions of all the detected species is available for the evaluation of using as a marker.

**Comment 16**. Please rewrite the conclusion.

Response: The conclusion part has been revised in the revised MS.

---

## Author Comment (AC5) · 5 Dec 2018

Reviewer #1:

(1) The introduction should be improved, to give more description of source profiles and its importance. Also, as a review paper, the developing history and shortages for current source profiles should be better summarized. The science implication should be highlighted.

Response:

This great point was also brought by reviewer #3. Our revision to this section is included in the following bulleted list:

1. Added sentences to the end of the first sentence of the second paragraph in the Introduction section to critical review the development of source profiles in China:

"The time evolution of source profiles is partly determined by the source apportionment techniques. In general, the receptor model was developed based on the assumption of mass conservation. A mass balance equation represents that the measured particle mass can be regarded as the linear sum of the mass of all chemical components contributed from several sources. Initially, the mass balance equations were deployed for a couple of specific elements and source types in America. Elements, ions and carbon materials gradually tend to be the routine chemical species in the source apportionment of PM. With the development of advanced sampling and chemical analysis techniques, more valuable information, such as organic compounds, isotopic measurement of radiocarbon, sulfur and nitrogen and high-resolution aerosol mass spectra and particle size distribution etc., have been explored to further expand the existing or new profiles. This information has been proved to provide source specificity capable of being incorporated into receptor models as new markers, constraining source contributions, and developing new models. For example, Dai et al (2019) developed a size-resolved CMB approach for source apportionment of PM based on the size profiles of sources. The new valuable information gives significant possibilities to source apportionment models to obtain more precise and reliable results.

2. Added a paragraph to review the current state of source profiles in China:

"Since the 1980s, source profile studies were initially implemented in China (Dai et al., 1987). During the past three decades, hundreds of source profiles have been achieved across China. These profiles covered more than forty cities and several source types. The main ubiquitous sources of atmospheric PM in China during the past three decades can be roughly divided into coal combustion sources (CC, with sub-type sources of coal-fired power plants, coal-fired boiler from industry and residential coal combustion), vehicle exhaust (VE, gasoline and diesel engines), industrial processes emissions (IE), biomass burning (BB), cooking emissions (CE), fugitive dust (FD, with sub-type sources of soil fugitive dust, construction dust and road dust) and other localized specific sources. These available profiles have filled the gap of the knowledge of source compositions and provided effective markers for the source apportionment studies. However, the current state and issues of pre-existing primary source profiles of PM in China are still unclear, it is time to overview these source profiles along the time line and add more profile knowledge to the atmospheric research community."

(2) As the introduction of a review articles, all related references should be added. For example, Line 72-75, references for organic compounds, isotope and size distribution should be all listed, not just listing some examples.

Response:

More references have been added to the related locations now. Thanks for your suggestion.

(3) The word evolution may be not suitable for the review of source profile. I believe change or variation is more suitable.

Response:

We disagree with the reviewer's comment that the word "evolution" is not suitable for the review of source profile, as our main point is to reveal the change of profiles along timeline. We appreciate the reviewer's reminder.

(4) The authors just use the source profile related keywords which may miss some

important papers. For example, you could not fine these key words in some tunnel or engine test studies. Also, the Elsevier database is not enough. Such as papers published on the journals of ACS, AGU, Springer will be missed.

Response:

This is another great point. We now have searched the literatures again based on a two-round paper search work and using more source-related key words. Details on the literature search of the main primary sources has been added to the Introduction section in response to this comment.

"To collect the potential published data related to source profiles, a two-round literature search work covering literature from 1980 to 2018 was done in this work. In the first round of searching, two authors are responsible for the same source to ensure every source category has been searched twice independently. The search keywords depend on source category. The following keywords for each source were used individually or in combination. As for CC sources, the key words are "coal combustion/coal burning/coal-fired boiler/coal-fired power plant/residential coal" and "source profile/chemical profile/particle composition". The key words for other sources are shown as follows. IE: "industrial emission" and "source profile/chemical profile/particle composition"; VE: "vehicle emission/exhaust emission/traffic emission/diesel engine/truck emission/gasoline engine/on-road vehicle/tunnel experiment/chassis dynamometer/portable emission measurement system" and "source profile/chemical profile/particle composition"; CE: "cooking emission" and "source profile/chemical profile/particle composition"; BB: "biomass burning/bio-fuel boiler" and "source profile/chemical profile/particle composition"; FD: "soil/fugitive dust/crustal material/construction dust/road dust" and "source profile/chemical profile/particle composition". Papers and dissertations in Chinese on China National Knowledge Infrastructure (CNKI) and papers in English on the web of science were searched using above keywords, respectively. The duplicated paper was then double-checked and excluded. The papers with topic related to source profiles but without providing any information of real-measured sources were also excluded. For example, papers reported source apportionment results with the use of PMF and CMB but without

reporting local profiles were not taken into account. As a result, a total of 193 papers have been collected from these efforts. In the second round of searching, the valid papers with available source profile data and detailed source sampling and chemical analysis methods were counted and used for post-analysis. Finally, a total of 456 published source profiles since the 1980s across China were collected."

(5) In the discussion section, more discussion should be added, not just say the higher or lower of components. Why they are higher or lower? For example, line 210-211.

Response:

Thanks for bring this comment to our attention. In the modified version, some discussions have been added to explain why some components are higher or lower. For example, in line 210-211 of old version as mentioned by the reviewer, more discussions have been added as follows:

"This difference was likely resulted from the combustion efficiency and desulfurization efficiency, as PPW was required to operate with high efficiency of desulfurization by the government while IBW was less under controlled."

(6) Line 131-132, the sentence indicated dilution sampling has been widely used, but the author just listed one paper. Li et al., 2009 is only for household biofuel burning test. There are many sentences have the same problem. That is, the author just listed one paper to say something. It is not suitable, especially for review articles. Such as Line 142-143, Line 179-183, Line 191-194.

Response:

Thanks for your comments. In the revised version, more references have been added in the updated MS to address this point. There are 90 new papers added in the updated version.

(7) In figure 2, change the medium volume sampling, there are also low-volume sampling methods used in source profile researches. Also in this figure, the sampling methods for vehicle emission should be given.

Response:

In the revised version, Figure 2 was modified according to this comment. The sampling methods of vehicle emission were given in the new Figure 2.

(8) Line 180, what is azzaarenes? It is a component or a type of components? Also the author use "a marker" which is false for plural. Same problem in Line 182.

Response:

Azzaarenes are nitrogen-heterocyclic polycyclic aromatic compounds. It is an organic component. "a marker" has been revised to "markers".

(9) Line 181, the references should be cited by year. The dot "ïijN" should be in English ","

Response:

Thanks. It has been revised.

(10) All the description about VOCs should be deleted in the paper.

Response:

In the revised version, all the VOCs parts have been deleted.

(11) Line 232-233, why wet desulfurization can cause the conversion of organics to OC?

Response:

The statement in the previous MS is not clear. In the revised version, the statement has been changed to 'OC in $PM_{2.5}$ profiles from the WFGD is also higher than that from DD, suggested that the possible conversion of gaseous or liquid organics to the particulate state in the lime slurry.' This statement is an inference. More investigation is needed in the future for addressing this point.

(12) In the discussion part, some sentences are not quantitatively. For example, Line 448- 449, the content of volatile components of the firewood is relatively high. The

authors should collect the data for volatile materials for different types of fuels and give more reliable results. Line 431-433, "much higher" indicated how much higher?

Response:

Thanks for your comments. In the revised version, the corrections have been made according to this comment. For the statement of 'the content of volatile components of the firewood is relatively high', we've checked the original reference of this point, and found it was just an inference. In the revised version, we have deleted this sentence.

(13) Line 390-391, how can the water-soluble ions contents itself suggests that insoluble matter is the main component? For many soluble components, the previous studies may not analyze them. The authors can only conclude which component are more soluble, but not for particles.

Response:

Thanks for your comment. The statement in our previous MS is not clear. This statement has been modified as 'In general, the total water–soluble ions only accounts for 0.0248-0.0648 g/g of fugitive dust.'

(14) Line 381-383, the author say Si is the predominant species, please give the mass percentages of Si in all the elements, not its content level. Similar description in other places.

Response:

Thanks. The statement in our previous MS is not clear enough. This statement has been modified as 'Si is the predominant species among the detected elements, accounting 42% mass of all the detected elements, followed by Fe, Na and Mg.'

(15) Line 367, Line 365, "generally higher", "relatively small", please give data;

Response:

Thanks for your comment. The previous statements were not clear enough. Some data values have been added in the revised MS.

(16) Line 363, their proportions were quite different, please give data;

Response:

The proportions contain different vehicle types in two countries with more than 20 data totally. These data could be found in the citation and Table S1 in the supplement materials.

(17) Line 351, I think it should be after 2011. Also, for the profiles, how can the authors know the source samples were just collected in 2005, 2008, and so on? Maybe the research published in 2011, but the samples were for older cars than 2008 or even 2005.

Response:

Thanks for your comment. We have checked the original literatures again, and confirmed the years that the samples were collected are correct.

(18) Line 442, different temperature between FCE and LCS, you mean the burning temperature or the sampling temperature. For the sampling test in LCS, dilution tunnel always reduced the high temperature flume gases to ambient temperature. I guess, it should be the Cl- depletion for ambient field sampling.

Response: For this point, we decide to delete the sentence to avoid misunderstanding of the comparison. More investigation is needed for addressing this point in the future.

The English should be improved and there are also obvious errors. I can just list some: (1) Line 79, the sentence should be corrected; (2) Line 304, "is" into "are".

Response: The English of the revised MS has been improved by a native speaker. We have checked the revised MS several times to correct the grammar errors.

---

## Author Response (AR2)

**Author's Response**

Dear Editor,

We sincerely thank all reviewers for their helpful comments and guidance. We have provided responses to Referee #4 comment below in blue.

Referee #4:

Almost all of my questions have been answered by authors, and I supposed that the manuscript could be published after some minor revisions.

1. The abstract should be rewritten and be more concise.

Response:

Thanks. We have rewritten and shortened the abstract (from 487 words to 300 words) in the revised MS to make it more concise as follows:

'Based on the published literatures and typical profiles from the source library of Nankai University, a total of 3326 chemical profiles of the main primary sources of ambient particulate matter across China from 1987 to 2017, are investigated and reviewed to trace the evolution of their main components and identify the main influencing factors to the evolution. In general, the source chemical profiles are varied with sources and influenced by different sampling methods. The most complicated profiles are likely attributed to coal combustion and industrial emissions. The profiles of vehicle emissions are dominated by organic carbon (OC) and elemental carbon (EC), and varied with the changing standards of sulfur and additives in the gasoline and diesel as well as the sampling methods. In addition to sampling methods, the profiles of biomass burning and cooking emissions are also impacted by the biofuel categories and cooking types, respectively. The variations of the chemical profiles of different sources, and the homogeneity of the sub-type source profiles within the same source category were examined with uncertainty analysis and cluster analysis. As a result, a relatively large variation has been found in the source profiles of coal combustion, vehicle emissions, industry emissions and biomass burning, indicating that these sources have the priority to establish the local profiles due to their high uncertainties. The presented results highlight the need for increasing investigation of more specific markers (e.g., isotopes, organic compounds and gaseous precursors) beyond routine measured components to discriminate sources. Although the chemical profiles of main sources have been reported previously in literatures, it should be noted that some of these chemical profiles are out of date currently, which needs to be updated immediately. Additionally, specific focus should be placed on the sub-type of source profiles in the future, especially for local industrial emissions in China.'

2. There still are some grammar faults, which should be corrected.

Response:

Thanks. We have checked the MS again and found some grammar faults, such as grammatical tense, singular & plural faults. All the corrections have been demonstrated in the revision mode of the revised MS followed this response.

3. The marks of the figures are not clear, especially for Figure 2.

Response:

Thanks. We have plotted the figures again in the revised MS to make them clearer.

[revised manuscript text omitted]

---

## Author Response (AR3)

**Author's Response**

Dear Editor,

We sincerely thank for your helpful comments and guidance. We have provided responses to your comment below in blue.

**Co-Editor's comments:**

Thanks for submitting the revised manuscript. It could be accepted for publication after addressing one remain issue. The data presented in the paper is valuable for the community and I strongly suggest to deposit data to a public data repositories and add a data availability statement in the manuscript. For your reference, please check the data policy of ACP.

https://www.atmospheric-chemistry-and-physics.net/about/data_policy.html

RESPONSE:

In the revised MS, we add the data availability statement as follows,

[revised manuscript text omitted]